# The interhemispheric CA1 circuit governs rapid generalisation but not fear memory

Heng Zhou[1,2], Gui-Jing Xiong[1], Liang Jing[1,3], Ning-Ning Song[4,10], De-Lin Pu[1], Xun Tang[2], Xiao-Bing He[5], Fu-Qiang Xu[5,6], Jing-Fei Huang[7], Ling-Jiang Li[8], Gal Richter-Levin[3], Rong-Rong Mao[1], Qi-Xin Zhou[1], Yu-Qiang Ding[4,8,10] & Lin Xu [1,2,6,8,9]

Encoding specificity theory predicts most effective recall by the original conditions at encoding, while generalization endows recall flexibly under circumstances which deviate from the originals. The CA1 regions have been implicated in memory and generalization but whether and which locally separated mechanisms are involved is not clear. We report here that fear memory is quickly formed, but generalization develops gradually over 24 h. Generalization but not fear memory is impaired by inhibiting ipsilateral (ips) or contralateral (con) CA1, and by optogenetic silencing of the ipsCA1 projections onto conCA1. By contrast, in vivo fEPSP recordings reveal that ipsCA1–conCA1 synaptic efficacy is increased with delay over 24 h when generalization is formed but it is unchanged if generalization is disrupted. Direct excitation of ipsCA1–conCA1 synapses using chemogenetic hM3Dq facilitates generalization formation. Thus, rapid generalization is an active process dependent on bilateral CA1 regions, and encoded by gradual synaptic learning in ipsCA1–conCA1 circuit.

[1] Key Laboratory of Animal Models and Human Disease Mechanisms, and Laboratory of Learning and Memory, Kunming Institute of Zoology, The Chinese Academy of Sciences, Kunming 650223, China. [2] School of Life Sciences, University of Science and Technology of China, Hefei 230027, China. [3] Department of Neurobiology and Ethology, and Department of Psychology, University of Haifa, Haifa 3498838, Israel. [4] Key Laboratory of Arrhythmias, Ministry of Education, East Hospital, and Department of Anatomy and Neurobiology, Collaborative Innovation Center for Brain Science, Tongji University School of Medicine, Shanghai 200092, China. [5] Wuhan Institute of Physics and Mathematics, The Chinese Academy of Sciences, Wuhan 430071, China. [6] CAS Centre for Excellence in Brain Science and Intelligent Technology, Shanghai 200031, China. [7] State Key Laboratory of Genetic Resources and Evolution, Kunming Institute of Zoology, The Chinese Academy of Sciences, Kunming 650223, China. [8] Mental Health Institute, Second Xiangya Hospital of Central South University, Changsha 410011, China. [9] KIZ-SU Joint Laboratory of Animal Model and Drug Development, College of Pharmaceutical Sciences, Soochow University, Suzhou 215123, China. [10] Present address: Institute of Brain Sciences, Fudan University, Shanghai 200031, China. Heng Zhou, Gui-Jing Xiong, Liang Jing and Ning-Ning Song contributed equally to this work. Correspondence and requests for materials should be addressed to Q.-X.Z. (email: qixin_zhou@vip.126.com) or to Y.-Q.D. (email: dingyuqiang@vip.163.com) or to L.X. (email: lxu@vip.163.com)

Encoding and later recall rarely recur in exactly the same conditions[1]. Recall flexibly across a variety of novel circumstances is largely dependent on generalization[2], a time-dependent capacity developed following the original memory. However, it is not fully understood how this capacity evolves from the original memories. This is also a primary question in the field of artificial intelligence regarding how to endow better generalization accuracy in dealing with unpredicted but similar circumstances that the "neural network" has never been trained on. In addition, overgeneralization of fear memories is one of the key symptoms in posttraumatic stress disorder (PTSD), which may occur with a passage of time after trauma. Thus, elucidating the neural basis of generalization holds the potential impact on these fields.

Research into the cellular and molecular mechanisms of memories have identified experience or activity-dependent fast changes of synaptic efficacy such as in CA3-CA1 synapses as one of the key mechanisms[3–5]. Theories of memory have proposed that the hippocampus learns quickly, and continuously interacts with the cortical systems to abstract gradually the common elements of memories for generalization[6–9], so as to endow the capacity of flexible recall while minimizing interference[10]. Optogenetic and chemogenetic tools enable studying in greater details the circuit mechanisms of memories. A recent study has demonstrated a global circuit mechanism by which the medial prefrontal cortex (mPFC) controls generalization of contextual fear memory during acquisition via the nucleus reuniens of the thalamus, which signals to the hippocampus and also back to the mPFC[11].

Considering from another perspective, we addressed here whether and how the hippocampal CA1 regions could process fear memory and generalization by recruiting separate mechanisms, and thereby minimizing interference of each another. In this study, we find a novel form of generalization with the characteristics of rapid development and easy extinction within 24 h after fear conditioning training. This rapid generalization but

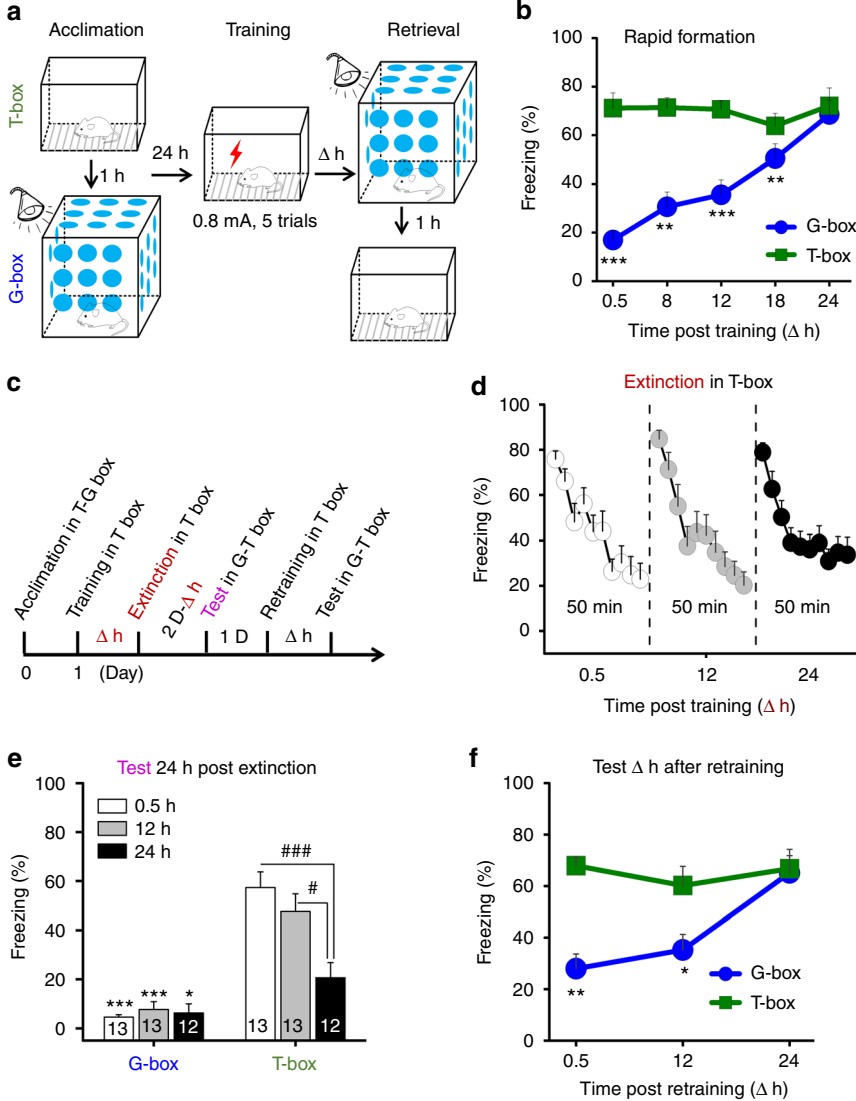

**Fig. 1** Formation and extinction characteristics of generalization. **a** Schematic of fear memory in the training box (T-box) and generalization in a non-training similar box (G-box), measured at Δ h after fear conditioning. **b** Generalization developed gradually in G-box over 24 h (n = 8 per group, but n = 9 at 18 h). **c** Generalization extinction paradigm. **d** Extinction curves in the separate groups by exposing into T-box for 50 min. **e** Generalization was disrupted in all groups, but fear memory was largely impaired at 24 h extinction after fear conditioning. **f** Retraining after extinction resulted in gradual reformation of generalization over 24 h (0.5 h, n = 6; 12 h, n = 7; 24 h, n = 8). Statistical comparisons are performed by using repeated two-way ANOVA (*contrast effects, G-box vs. T-box; #parameter estimates, between T-box groups); *P < 0.05, **P < 0.01, *** or ### P < 0.001. Error bars, s.e.m

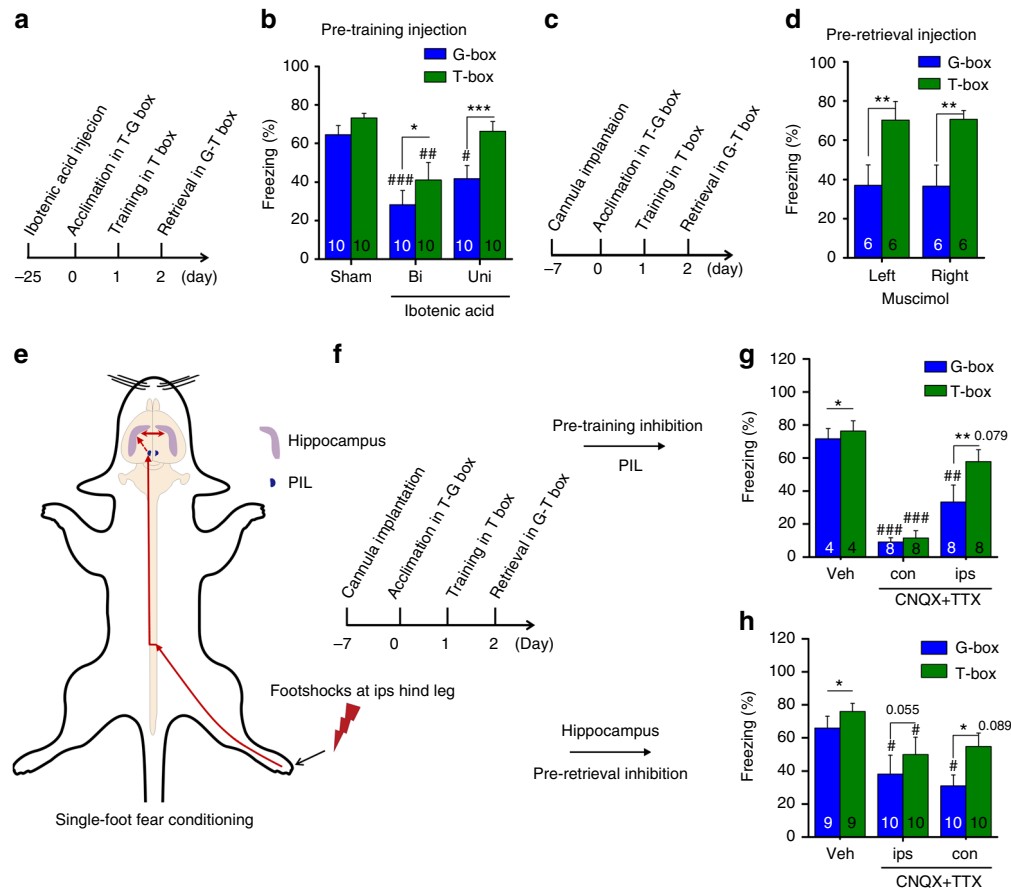

**Fig. 2** Bilateral CA1 are required symmetrically for generalization. **a** Paradigm for studying CA1 lesion effects by using ibotenic acid. **b** Unilateral (Uni) CA1 lesion impaired generalization, but bilateral (Bi) CA1 lesion impaired both fear memory and generalization relative to Sham control. **c** Paradigm for studying inhibition effects of CA1. **d** Inhibition of left or right CA1 before retrieval tests by using muscimol impaired generalization near equally. **e**, **f** Schematic of single-foot fear conditioning and studying paradigm. **g** Inhibition of ipsPIL before fear conditioning by using CNQX + TTX impaired generalization, with nonsignificant effect on fear memory relative to Veh, but that inhibiting conPIL impaired both fear memory and generalization. **h** Using this single-foot paradigm, however, inhibition of ipsCA1 or conCA1 before retrieval tests impaired generalization similarly, while ipsCA1 or conCA1 inhibition caused nonsignificant reduction of fear memory relative to Veh. Statistical comparisons are performed by using two-way ANOVA (*contrast effects for G-box vs. T-box; #parameter estimates for control vs. treatment); * or #$P < 0.05$, ** or ##$P < 0.01$, *** or ###$P < 0.001$. Error bars, s.e.m. PIL, the posterior intralaminar nuclei of the thalamus. Veh, vehicle; ips, ipsilateral; con, contralateral

not fear memory is processed by the interhemispheric exchanges of the fear memory information allocated in bilateral CA1, leading to a gradual developing synaptic potentiation in the ipsCA1–conCA1 circuit within 24 h.

## Results

**Formation and extinction of generalization**. We measured generalization and contextual fear memory by placing the conditioned rats into a non-training similar box (G-box, blue) and 1 h later into the training box (T-box, green) (Fig. 1a), at 0.5, 8, 12, 18, and 24 h after fear conditioning for the separate groups (Fig. 1b, Δ h). As shown in Fig. 1b, fear memory was formed quickly in T-box, and maintained over the retrieval tests. By contrast, generalization was developed gradually in G-box, and rapidly rose up to maximal over 24 h, when recall efficiency became near equivalent to that in T-box (Fig. 1b, time×box interaction, $F_{(4,36)} = 3.575$, $P = 0.015$, two-way ANOVA; G-box, *contrast effects between T-box vs. G-box at Δ h following $F_{(1, 36)} = 110.14$, $P < 0.001$, repeated two-way ANOVA), demonstrating a rapid time-course of generalization full formation distinct from that of fear memory. This is a novel form of generalization likely differed from the more gradual one reported

previously showing that several weeks are required for generalization full development in mice[12,13].

To explore the characteristics of this novel generalization in more details, which might be useful for studying its underlying mechanism, we measured the generalization and fear memory 24 h after extinction training that was performed by exposing the rats into T-box for 50 min (Fig. 1c), starting at 0.5, 12 or 24 h after fear conditioning for the separate groups (Fig. 1d, Δ h). These time points represented increasing levels (low to high) of generalization that was then maintained for at least 7 d (see Veh groups in Supplementary Fig. 2e, f). Consistent with previous report[14], extinction training at the early stages (0.5 and 12 h) had nonsignificant effect on fear memory, but that at the later time point (24 h) largely reduced fear memory (i.e., suppression) (Fig. 1e, time×box interaction, $F_{(2,35)} = 12.090$, $P < 0.001$, two-way ANOVA; T-box: 0.5 h vs. 24 h, ###$P < 0.001$, and 12 h vs. 24 h, #$P < 0.03$; parameter estimates, repeated two-way ANOVA). In contrast, extinction training at all these stages (0.5, 12, and 24 h) disrupted the generalization (i.e., renewal) (Fig. 1e, G-box: T-box vs. G-box at the time points, *contrast effects, repeated two-way ANOVA). This revealed also a distinct pattern of rapid generalization in memory extinction. Retraining after extinction caused gradual generalization reformation over 24 h

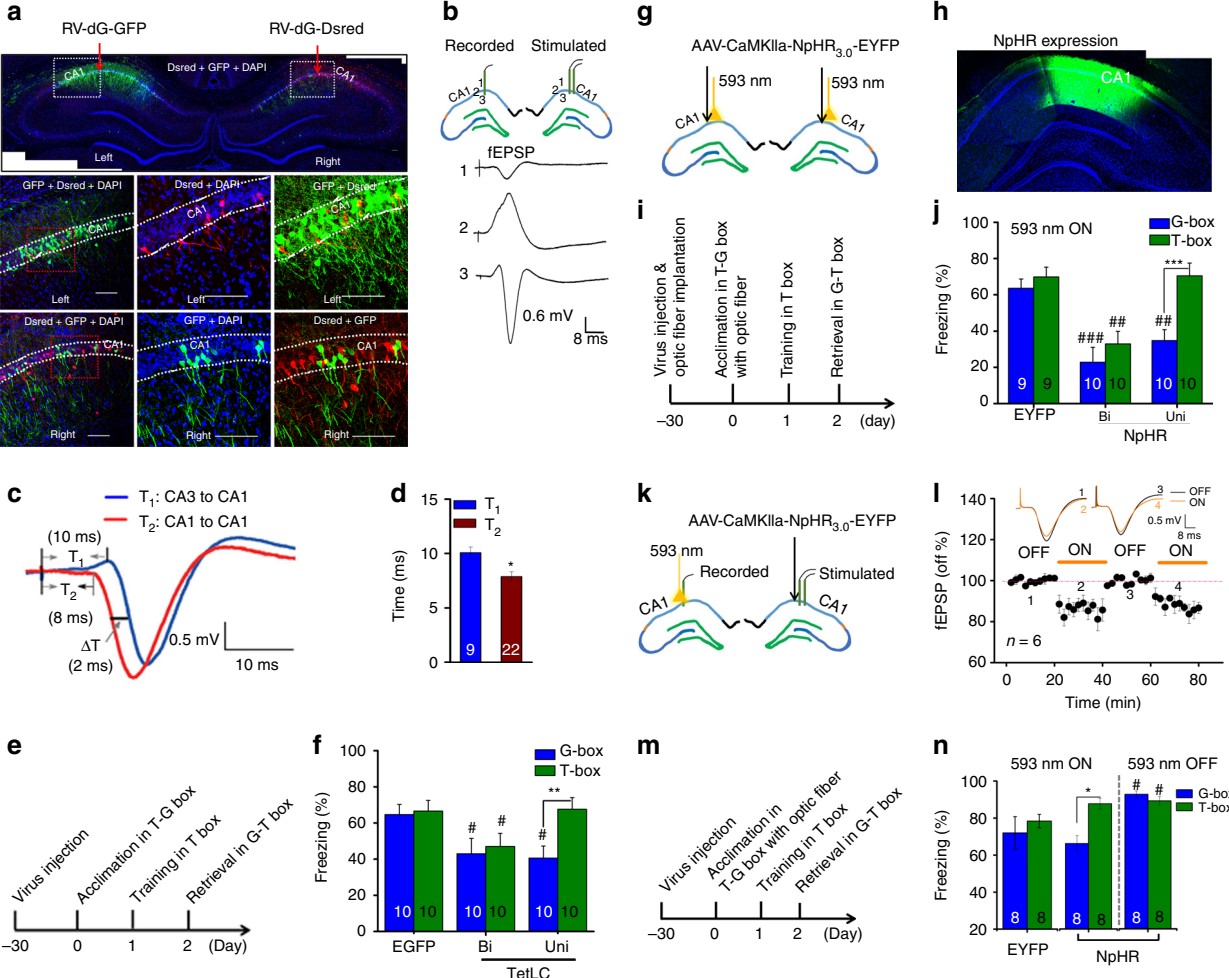

**Fig. 3** The ipsCA–conCA1 functional connectivity. **a** RV-dG-GFP and -Dsred were infused into the stratum oriens of left and right CA1, and a number of neurons were labeled in the opposite side. **b** In vivo study revealed that electrical stimulation at ipsCA1 evoked the excitatory postsynaptic potentials (fEPSP) in conCA1, with an onset latency of about 8-ms (**c**, **d** T2), relative to about 10-ms in the CA3-CA1 pathway (**c**, **d** T1). **e** Paradigm for studying TetLC effects. **f** Unilateral (Uni) or bilateral (Bi) CA1 expression of TetLC (AAV-Syn-EGFP-2A-TetLC-3flg) impaired generalization or the both, respectively. **g**–**j** NpHR was expressed and optogenetic stimulation was used during the retrieval tests. Silencing of Uni or Bi CA1 impaired generalization or the both, respectively. **k**–**n** When NpHR was expressed at ipsCA1 but optogenetic stimulation was applied at conCA1 to silence the ipsCA1 projection terminals onto conCA1, the ipsCA1–conCA1 fEPSP was reduced by about 20% with light on relative to light off. Light on during the retrieval tests impaired generalization, while light off during the tests had a slight enhancement effect on the both. Statistical comparisons are performed by using one-tail $t$-test **d** and two-way ANOVA (**f**, **j**, **n** *contrast effects; #parameter estimates); * or #$P < 0.05$, ** or ##$P < 0.01$, *** or ###$P < 0.001$. Error bars, s.e.m. EGFP or EYFP, control virus; Calibration bar: 100 μm

again (Fig. 1f, time×box interaction, F $(2,18) = 18.551$, $P < 0.001$, two-way ANOVA; *contrast effects between T-box vs. G-box at $\Delta$ h following F $(1,18) = 71.932$, $P < 0.001$, repeated two-way ANOVA).

Accordingly, we propose that the circuit mechanism for rapid generalization is separated from that of fear memory, regardless of whether or not they share the similar cellular and molecular mechanisms.

## Generalization requires bilateral CA1

We then started to explore the circuit mechanism of the rapid generalization using pharmacological tools. Ibotenic acid was used to lesion the CA1 region(s) of the dorsal hippocampus (Supplementary Fig. 1) 25 d before fear conditioning (Fig. 2a). We found that unilateral CA1 lesion might impair generalization (Fig. 2b, group×box interaction, F $(2,34) = 2.455$, $P = 0.101$; within groups or between boxes, F $(2,34) = 7.933$, $P = 0.001$ or F $(1,34) = 29.66$, $P < 0.001$; two-way

ANOVA; Uni: #$P = 0.02$, G-box vs. Sham, parameter estimates, two-way ANOVA; ***$P < 0.001$, G-box vs. T-box, contrast effects, two-way ANOVA) without affecting fear memory (Fig. 2b, Uni: $P = 0.403$, T-box vs. Sham, parameter estimates, two-way ANOVA), while bilateral CA1 lesion might impair both fear memory and generalization (Fig. 2b, Bi: ###$P < 0.001$, G-box vs. Sham; ##$P = 0.002$, T-box vs. Sham, parameter estimates, two-way ANOVA; *$P = 0.034$, G-box vs. T-box, contrast effects, two-way ANOVA), examined 24 h after fear conditioning.

A set of experiments was then performed to confirm this intriguing finding by using the AMPA receptor inhibitor CNQX plus the sodium channel blocker TTX (CNQX + TTX), the GABA$_A$ receptor agonist muscimol, and the protein synthesis inhibitor anisomycin, which is known to impair long-term contextual fear memory[15]. Unilateral or bilateral CA1 inhibition using CNQX + TTX before fear conditioning (Supplementary Fig. 2a, b) or before retrieval test (Supplementary Fig. 2c, d, CNQX + TTX) resulted in similar effects on generalization,

examined 24 h after fear conditioning. Using muscimol before retrieval test to inhibit unilateral or bilateral CA1 also led to similar effects (Supplementary Fig. 2c, d, Muscimol). The drug effects on generalization were long-lasting, because either CNQX + TTX or anisomycin infused into unilateral or bilateral CA1 after fear conditioning impaired generalization or the both, tested 7 d after fear conditioning (Supplementary Fig. 2e, f). Thus, generalization appears to be a type of memory not only depending on but also differing from the original memory.

We also examined whether left and right CA1 regions were (a) symmetrically required for generalization, i.e., either side of the CA1 was near equivalently important for generalization. Injection of muscimol to inhibit left or right CA1 before retrieval test led to nearly identical results, impairing generalization (Fig. 2c, d, laterality×box interaction, $F_{(1,10)} = 0.003$, $P = 0.957$; $F_{(1, 10)} = 19.656$, **$P = 0.001$ for G-box vs. T-box; but $P = 0.998$ or $0.957$ for left vs. right or box vs. laterality; two-way ANOVA). Consistent with a recent report indicating that unilateral lesion of the dorsal hippocampus has no effects on contextual fear memory[16], our findings indicate that unilateral CA1 suffices for fear memory. However, generalization requires bilateral CA1 activity symmetrically, implicating that the interhemispheric exchange of memory information through the dorsal hippocampal commissure (DHC) is critical. This is potentially interesting because the DHC is highly conserved in mammals and believed to present a functional pathway also in humans[17,18].

To explore further whether bilateral CA1 activity is indeed symmetrical for generalization, we applied a single-foot fear conditioning paradigm by giving footshocks directly to the right hind leg of the rats (Fig. 2e, f). It is documented that spinal cord neurons relay the signals of the footshocks to the posterior intralaminar nuclei (PIL) of the thalamus predominantly *via* contralateral projections[19,20], and then to the hippocampus[21]. Therefore, we examined whether and how PIL inhibition could affect fear memory and/or generalization. We found that inhibition of conPIL using CNQX + TTX before single-foot fear conditioning impaired both fear memory and generalization (Fig. 2g, group×box interaction, $F_{(2, 17)} = 7.065$, $P = 0.006$; both T-box and G-box, ###$P < 0.001$, Veh vs. con, parameter estimates, two-way ANOVA), but that inhibiting ipsPIL impaired generalization (Fig. 2g, G-box, ##$P < 0.006$, Veh vs. ips, parameter estimates, two-way ANOVA) with smaller nonsignificant effect on fear memory (Fig. 2g, T-box, $P = 0.079$, Veh vs. ips, parameter estimates, two-way ANOVA). In marked contrast, inhibition of ipsCA1 or conCA1 using CNQX + TTX after single-foot fear conditioning (Supplementary Fig. 3a, b) or before retrieval test similarly impaired generalization (Fig. 2h, group×box interaction, $F_{(1,26)} = 1.439$, $P = 0.255$; G-box, #$P = 0.035$ or $0.01$, Veh vs. ips or con, parameter estimates; ips or con, $P = 0.055$ or *$P = 0.018$, G-box vs. T-box, contrast effects; two-way ANOVA). Fear memory was also affected in a milder extent (Fig. 2h, T-box, #$P = 0.039$, Veh vs. ips; $P = 0.089$, Veh vs. con; parameter estimates, two-way ANOVA). Thus, these findings indicate that such asymmetry in the PIL, conPIL responsible for both fear memory and generalization while ipsPIL responsible for generalization, is somehow emerged into a symmetry in which bilateral CA1 is responsible for generalization while unilateral CA1 is sufficient for fear memory.

**The ipsCA1–conCA1 connectivity**. The above studies have identified that generalization depends on bilateral CA1 activity, and requires symmetrical ipsCA1 and conCA1 activity even in single-foot fear conditioning. The DHC could provide a route for interhemispheric exchanges between ipsCA1 and conCA1 to lead

to such symmetry in CA1. However, this pathway was reported to have only weak bilateral connections which vary along the septo-temporal axis of the hippocampus[22]. We then set out to reexamine CA1 bilateral connectivity using rabies virus (RV-dG-GFP and −Dsred, non-transsynaptic tracing virus) and electro-physiological methods.

RV-dG-Dsred was injected into ipsCA1 stratum oriens, and a number of neurons were found to be labeled in conCA1 (Supplementary Fig. 4), suggesting that the labeled neurons have projection terminals onto ipsCA1 stratum oriens. RV-dG-GFP and −Dsred were injected into stratum oriens of ipsCA1 and conCA1 respectively, and again, a few neurons were labeled in the opposite side (Fig. 3a, and Supplementary Fig. 5), also indicating that the ipsCA1−conCA1 projections might directly cross the midline below the corpus callosum.

When RV-dG-GFP was directly infused into the midline, adequate connections between ipsCA1 and conCA1 were revealed with the labeled neurons and fibers exclusively covering bilateral the whole CA1 regions (Supplementary Fig. 6). By marked contrast, RV-dG-GFP was injected into ipsCA1 stratum radiatum, and multiple neurons were labeled in both ipsCA3 and conCA3 but not in conCA1 (Supplementary Fig. 7). Thus, it is possible that ipsCA1 receives bilateral CA3 inputs mainly at stratum radiatum but it can sense conCA1 inputs mainly at the stratum oriens. Consistent with these findings, ample projection terminals were also found in conCA1 stratum oriens after infusion of AAV-CaMKIIa-ChR2-EYFP into ipsCA1 (Supplementary Fig. 8a), at which optogenetic stimulation effectively evoked the field excitatory postsynaptic potentials (fEPSP) in conCA1 (Supplementary Fig. 8b).

Both in vitro and in vivo fEPSP recordings were further used to characterize the ipsCA1−conCA1 functional connectivity. Electric stimulating at the midline below the corpus callosum effectively evoked a fEPSP in both ispCA1 and conCA1 in a coronal slice (Supplementary Fig. 6d, e), indicating the ipsCA1−conCA1 functional connectivity in the septal (dorsal) part of the hippocampus.

In vivo recording revealed that the fEPSP at conCA1 was effectively evoked by stimulating at ipsCA1 using homotopic coordinates (Fig. 3b), with an onset latency of about 8-ms, compared with about 10-ms in the ipsCA3-conCA1 synaptic transmission (Fig. 3c, d, *$P = 0.01$, T1 vs. T2; one-tail $t$-test). To rule out the possibility that the stimulation antidromically excited ipsCA3 which in turn drove the ipsCA3-conCA1 fEPSP, although the onset latency is presumably even longer, we used CNQX + TTX. The ipsCA1−conCA1 fEPSP was blocked by infusion of CNQX + TTX into ipsCA1, but not into ipsCA3 (Supplementary Fig. 8c, d).

Together, the results indicate that the ipsCA1−conCA1 functional connectivity may serve as a repository that may sense the signals of bilateral CA1 activities, whereas its functions have remained not clear.

To establish whether ipsCA1−conCA1 functional connectivity is critical for generalization but not for fear memory, a set of experiments was then performed. First, we injected AAV-syn-EGFP-2A-TetLC into CA1 30 d before fear conditioning (Fig. 3e) to express tetanus toxin light chain (TetLC) that is known able to block neurotransmitter release at the CA1 efferent projection terminals. Unilateral CA1 expression of TetLC impaired generalization (Fig. 3f, group×box interaction, $F_{(2,27)} = 9.322$, $P = 0.001$; T-box $P = 0.915$, G-box #$P = 0.037$, EGFP vs. Uni, parameter estimates; T-box vs. G-box, **$P = 0.003$, contrast effects; two-way ANOVA), but bilateral CA1 expression of TetLC impaired the both (Fig. 3f, T-box #$P = 0.043$, G-box #$P = 0.022$, EGFP vs. Bi, parameter estimates; T-box vs. G-box, $P = 0.265$, contrast effects; two-way ANOVA; see also Supplementary Fig. 9), relative to

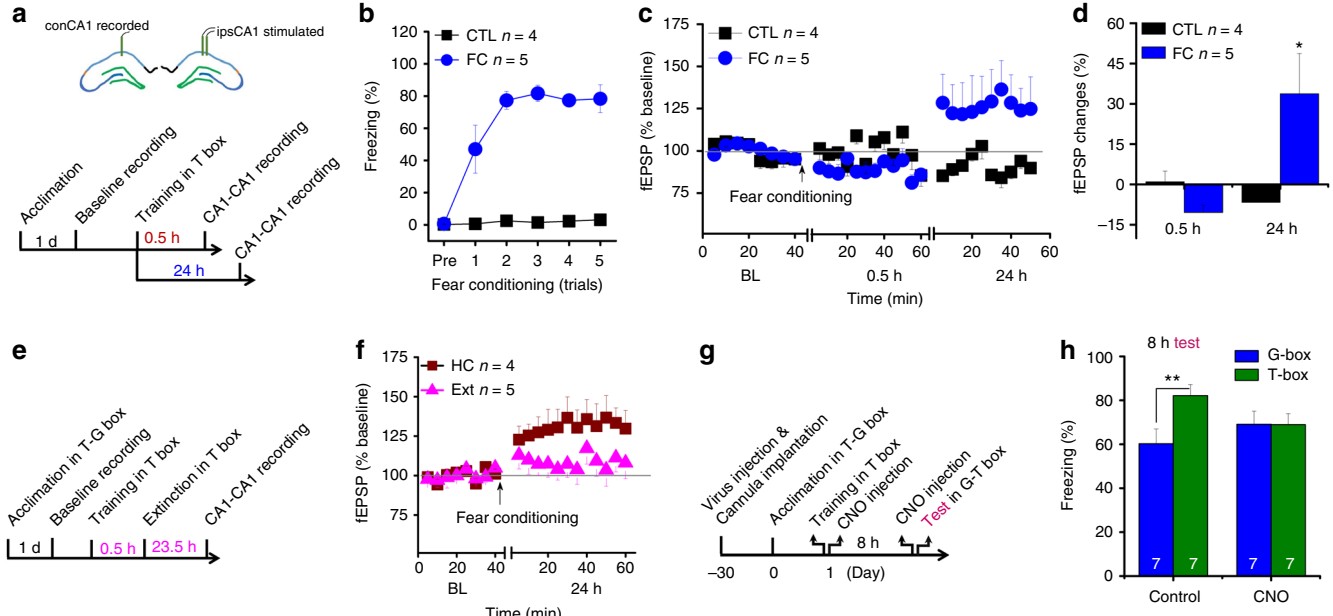

**Fig. 4** The ipsCA1–conCA1 synaptic efficacy is readjusted selectively for generalization. **a**, **b** After 40-min baseline recordings for ispCA1-conCA1 fEPSP, animals were subjected to fear conditioning (FC). Control animals (CTL) were exposed to the training box without footshocks. **c**, **d** No difference from baseline (BL) was detected at 0.5 h after FC, but a reliable synaptic potentiation was found to be developed over 24 h after FC when generalization was well developed. **e**, **f** No difference from BL was found at 24 h after FC if extinction training (Ext) was applied at 0.5 h to disrupt generalization formation, while a synaptic potentiation at 24 h after FC was observed in homecage animals (HC) without Ext. **g**, **h** AAV-hsyn-hM3Dq-mCitrine was injected into ipsCA1, while guide cannula was implanted into conCA1. Application of clozapine-N-oxide (CNO group) twice, immediately after fear conditioning and 30 min before retrieval tests at 8 h via the guide cannula, enhanced generalization in a similar box (G-box), up to the levels almost equivalent to that in the training box (T-box). Without CNO treatment (saline control), the scores in G-box were significantly lower than that in T-box. Statistical comparisons are performed by using repeated one-way ANOVA **d**, **f** or two-way ANOVA **h**; *$P < 0.05$, **$P < 0.01$. Error bars, s.e.m. ipsCA1, ipsilateral CA1; conCA1, contralateral CA1

control virus (Fig. 3f, EGFP), suggesting that the CA1 efferent projections are responsible for generalization.

Second, direct silencing unilateral or bilateral CA1 neurons expressed optogenetic AAV-CaMKIIa-NpHR$_{3.0}$-EYFP during the retrieval tests (Fig. 3g–i) produced the similar results (Fig. 3j, group×box interaction, $F_{(2,26)} = 16.261$, $P < 0.001$; T-box, $P = 0.95$, EYFP vs. Uni; ##$P = 0.001$, EYFP vs. Bi; G-box, ##$P < 0.006$, EYFP vs. Uni; ###$P < 0.001$, EYFP vs. Bi; parameter estimates; Uni or Bi, ***$P < 0.001$ or *$P = 0.04$, T-box vs. G-box; contrast effects; two-way ANOVA), relative to control virus (AAV-CaMKIIa-EYFP) (Fig. 3j, EYFP).

Finally, AAV-CaMKIIa-NpHR$_{3.0}$-EYFP was injected into ipsCA1, and optogenetic stimulation was applied at the projection terminals onto conCA1, by which the ipsCA1–conCA1 fEPSP was reduced by about 20% with light on relative to light off (Fig. 3k, l). Generalization might be impaired by turned on optogenetic stimulation (group×box interaction, $F_{(2,21)} = 3.309$, $P = 0.384$; within groups or between boxes, $F_{(2, 21)} = 6.836$, $P = 0.005$ or $F_{(1, 21)} = 4.918$, $P = 0.038$; *$P = 0.017$, T-box vs. G-box, contrast effects, two-way ANOVA) with nonsignificant effect on fear memory ($P = 0.283$, EYPF vs. T-box, parameter estimates, two-way ANOVA) during retrieval tests; a slight enhancement of both fear memory and generalization was observed by turned off the stimulation relative to EYFP control (Fig. 3m, n, both T-box and G-box, #$P = 0.022$, EYPF vs. light off, parameter estimates, two-way ANOVA). Altogether, activities of the ipsCA1–conCA1 projections were found to be particularly essential for generalization.

**The ipsCA1–conCA1 circuit for generalization**. To establish further how the ipsCA1–conCA1 circuit contributes to

generalization, we recorded experience-dependent changes of synaptic efficacy in the ipsCA1–conCA1 circuit in freely moving rats. The fEPSP baseline (BL) was recorded for 40-min before fear conditioning (Fig. 4a, b), and then the fEPSP was recorded for 1 h starting at 0.5 h after fear conditioning. No difference from BL was detected at 0.5 h after fear conditioning (Fig. 4c, d), when generalization was nearly undetectable (see Fig. 1a, b, 0.5 h). By contrast, the fEPSP recorded at 24 h after fear conditioning exhibited a reliable synaptic potentiation relative to BL (Fig. 4c, d, 24 h, $F_{(1, 7)} = 5.623$, *$P = 0.049$, repeated ANOVA), when generalization was well developed (see Fig. 1a, b, 24 h).

Furthermore, our behavioral results indicated that extinction training starting at 0.5 h after fear conditioning disrupted generalization without apparent effects on fear memory, tested 24 h later (see Fig. 1c–e). We thus used this paradigm to examine whether a selective disruption of generalization would readjust this late developing synaptic potentiation. The fEPSP recorded for 1 h starting at 23.5 h after the extinction training was not different from BL (Fig. 4e, f), when generalization was disrupted by the extinction paradigm. A late developing synaptic potentiation was observed again if the conditioned animals was left in homecage without extinction training (Fig. 4f, $F_{(1, 7)} = 6.653$, $P = 0.037$, repeated ANOVA).

Moreover, if generalization and gradual developing synaptic potentiation were causally linked together, direct exciting of the ipsCA1–conCA1 synapses may speed up the active process for generalization formation. To excite these synapses continuously for hours that might overlay on the endogenous active process of the selected neurons in developing a synaptic potentiation gradually, we used a chemogenetic tool, AAV-hsyn-hM3Dq-mCitrine that was injected into ipsCA1 (Fig. 4g, and

Supplementary Fig. 10). Injection of clozapine-N-oxide (CNO) twice, immediately after fear conditioning and 30 min before retrieval tests at 8 h, into conCA1 excited the postsynaptic neurons that had the contralateral inputs from the ipsCA1 neurons directly (as indicated by cFos expression, see Supplementary Fig. 10). We found that generalization was facilitated, up to maximal levels already at 8 h after fear conditioning (Fig. 4h, treatment×box interaction, F (1, 12) = 8.527, P = 0.013; CNO, P = 0.983, T-box vs. G-box, contrast effects, two-way ANOVA), almost equivalent to that in T-box. By contrast, without CNO treatment, generalization was significantly lower than that in T-box (Fig. 4h, control, **P = 0.002, T-box vs. G-box, contrast effects, two-way ANOVA). These findings indicate that indeed, gradual developing synaptic potentiation of the ipsCA1–conCA1 circuit is a critical part for the development of generalization.

## Discussion

The hippocampal CA1 regions have been implicated in both memory and generalization, which raises the question of how these two tasks function alongside each other without interference. Using non-transsynaptic rabies virus, we find that CA3 neurons send the Schaffer and commissural projections mainly onto synapses in stratum radiatum of ipsCA1 and conCA1, but an ipsCA1–conCA1 circuit is consisted of the commissural projections onto synapses mainly in CA1 stratum oriens. Notably, we demonstrate here for the first time that generalization is dependent on symmetrical ipsCA1–conCA1 activity, and maintained by gradual developing potentiation of synaptic efficacy in the ipsCA1–conCA1 circuit. By contrast, fear memory is widely believed to involve fast potentiation of synaptic efficacy in the synapses from CA3 to CA1 stratum radiatum[3–5]. Therefore, these spatiotemporally separated mechanisms within CA1 at least in the early stage of memory processing, may match the orthogonal properties for generalization theoretical hypothesis described previously[1], thereby minimizing interference of one to the other[10]. Furthermore, it is apparent not because fear memory is not specific or precise at this early stage while resistant to extinction training (see Fig. 1e, T-box), but because generalization is gradual developing and thus sensitive to extinction training (see Fig. 1e, G-box).

A number of studies have demonstrated gradual generalization, for which 14 d[23–26] or even longer[12,13,27,28] are required for its full formation, for which the freezing level in G-box becomes near equivalent to that in T-box. Theories have been developed for understanding this time-dependent process of generalization that may relate to systems consolidation of memory when the contextual components may become less specific[13,27]. In contrast, early but not full generalization within 2 d after fear conditioning can occur in mice by the learning process in a neural circuit that involves the mPFC, the thalamus, and the hippocampus[11]. Furthermore, full generalization may have occurred within 24 h if the rats were treated with D-cycloserine, an enhancer of NMDA receptor activity, before one-trial inhibitory avoidance training[26]. This is somewhat consistent with our present finding in which generalization was fully developed within 24 h, likely due to pre-exposure to T-box and 1 h later to G-box on the acclimation day, by which NMDA receptors would have been activated, leading to the formation of the context memories 24 h before fear conditioning. Moreover, the context memories for T-box and G-box could be linked together as a result of the temporally closed neuronal ensembles, as suggested by a recent report and the "allocate to link" hypothesis[29,30]. It seems that the linked context memories would allow fear to be transferred (or generalized) from T-box to G-box rapidly within 24 h. Consistent with this assumption, without such a memory link by using the protocols

similar as previous generalization studies described through exposing the animals to T-box but not G-box on the acclimation day, full development of generalization in G-box was observed by two weeks after fear conditioning (Supplementary Fig. 11). All together, we would like to propose a hypothesis for which the formation of generalization is gradual but have actively rapid and passively slow phases, likely corresponding to cellular consolidation and systems consolidation of memory when the contextual components are more or less specific. This explanation seems consistent with clinical recommendation that exposure-based therapy early after trauma in susceptible individuals may produce better efficacy in reducing overgeneralization of conditioned fear and prevalence of PTSD[31,32].

Fear memory appears to be processed by contralateral side of the subcortical systems such as the amygdala[33] or the thalamic PIL (see Fig. 2e–g), if used lateralization of the stimulation during fear conditioning. In marked contrast, fear memory is likely duplicated in bilateral CA1 regions so that either CA1 is sufficient for specific recall, but generalization requires symmetrical activity of bilateral CA1. Their activity history is likely to explain how bilateral CA1 neurons are selected for developing generalization as suggested recently[29,30], for example, hippocampal activities can be synchronized at times between ipsCA1 and conCA1[34]. Thus, the synchronized activities at the early stage could have readjusted the ipsCA1–conCA1 functional connectivity in the selected bilateral CA1 neurons, presumably through spike timing-dependent plasticity mechanisms[35–37], to enable effective generalization formation at the rapid phase.

Specificity of memory is proposed to be maintained by "pattern separation" through the DG-CA3 circuit mechanisms including DG adult neurogenesis over time[38–40], while generalization is suggested to occur at recall through "pattern completion" in the CA3 recurrent system[7,38]. Given the present findings, the distinct processes for generalization are expected to involve also inter-hemispheric exchanged activities, thereby contributing to different aspects of memory and generalization through spatiotemporally separated dimensions. These intra-hippocampal processes should be able to interact continuously with the cortical and subcortical systems to enable recall more flexibly[6–9,11]. Because generalization and related plasticity were found to be developed gradually over time, here rapid generalization is likely to present an example of a process that requires "gradual internal learning", as proposed by multiple theories of memory[6–9]. We here propose that "gradual internal learning" is continuously readjusting the functional connectivity of the neural circuits not only for minimizing interference but also for reducing unpredicted errors under varying circumstances. Our findings suggest that rapid generalization requires "gradual internal learning" over 24 h, a process distinct from that of memory itself, which is dependent on bilateral CA1 activities to readjust the interhemispheric CA1–CA1 synaptic efficacy for the selective formation and also extinction of generalization.

## Methods

**Subjects**. Male Sprague-Dawley rats (from Animal Center of Kunming Medical University, Yunnan, China) and C57 mice (from Vital River Laboratory Animal Technology Co. Ltd., Beijing, China), aged 10–12 weeks, were used. All experiments were carried out in rats, except those reported in Supplementary Figs. 4–7 in which mice were used. Animals were group housed in ventilated cages with free access to water and food, a 12/12-h light/dark cycle, and a thermoregulated environment. As usually, animals were randomly grouped, and more than 7 animals per group were used for behavioral study but 4–5 animals per group for electrophysiological study. None of the animals finished experimental design was excluded from analyses. Most of the experiments were performed by the equal contribution authors who were not blinded to the experimental groups, but each of they performed independent experiments. Experimental protocols were approved by the animal ethics committee of Kunming Institute of Zoology, Chinese Academy of Sciences.

**Surgery, cannula implantation and chemicals**. Using the techniques described previously[41–49], surgery was carried out under pentobarbital sodium (Sigma, 60 mg kg$^{-1}$, i.p.) anesthesia, medical oxygen (95% $O_2$ and 5% $CO_2$) was supplied through a mask, and body temperature (37 ± 0.5 °C) was maintained through a heating pad. After surgery, animals were individual housed for at least 7 d before experiments or mentioned elsewhere.

Under the surgery conditions, stainless steel guide cannulas (26 gauges) were implanted by using stereotaxic apparatus (RWD Life Sciences, Shenzhen, China), according to the Paxinos and Watson Brain Atlas. Guide cannulas were designed 1 mm shorter than injection needles, and placed by 1 mm above the targeted areas, the CA1 regions of the dorsal hippocampus using the stereotaxic coordinates: anteroposterior (AP), −3.8 mm, mediolateral (ML) ± 2.8 mm, and dorsoventral (DV) −2.6 mm. The stereotaxic coordinates for the posterior intralaminar nuclei (PIL) of the thalamus are AP, −5.6 mm, ML ± 2.9 mm, and DV −6.8 mm. The guide cannulas were plugged with stylets, and affixed to the skull of the rats using dental cement with two stainless steel screws serving as anchors.

The AMPA receptor inhibitor CNQX, the sodium channel blocker TTX, the GABAA receptor agonist muscimol, the protein synthesis inhibitor anisomycin, and ibotenic acid were all purchased from Sigma. Muscimol (4.38 mMol L$^{-1}$)[50], CNQX (3 mMol L$^{-1}$) + TTX (20 μMol L$^{-1}$)[51], and anisomycin (0.3 mMol L$^{-1}$)[46] were all infused at a volume of 1 μL per side through the implanted guide cannulas by using the injection needles connecting to a syringe pump (RWD Life Sciences, Shenzhen, China), and infused at a speed of 0.1 μL min$^{-1}$.

**Lesion of CA1 regions**. Experimental procedures were used similar to those previously described[52]. Under the surgery conditions, injection needles were positioned to the actual coordinates (AP, −3.8 mm, ML ± 2.8 mm, DV −2.6 mm) for the CA1 regions of the dorsal hippocampus using stereotaxic apparatus, without implantation of the guide cannulas. The injection needles, connected to a syringe pump, delivered ibotenic acid (Sigma, 1 μL, 1.5 μg, dissolved in sterile saline) into unilateral (Uni) or bilateral (Bi) CA1 regions. The animals were individual housed for 25 d before experiments. After experiments, the lesion site was confirmed by using histological method (see Supplementary Fig. 1).

**Viral tracing**. Using the techniques described previously[53], non-trans-synaptic rabies viruses (RV-dG-Dsred and RV-dG-GFP, titer ~ 10$^8$ mL$^{-1}$, Brain VTA Inc., Wuhan, China) were used to trace ipsilateral (ips) CA1-contralateral (con) CA1 projections. Under the surgery conditions, glass micropipettes were positioned to the targeted areas using stereotaxic apparatus: the stratum oriens or stratum radiatum of the CA1 regions in the dorsal hippocampus (the rat: AP, −3.8 mm, ML ± 2.8 mm, DV −2.6 mm; the mouse: AP = −2.0 mm, ML = −1.3 mm, DV = −1.25 mm for the stratum oriens of CA1 or −1.35 mm for the stratum radiatum of CA1), and the midline of the dorsal hippocampal commissure (DHC) below corpus callosum (the mouse: AP = −0.9 mm, ML = 0 mm, DV = −2.0 mm). Infusion of the viruses (1 μL in CA1 and 0.3 μL in midline) using glass micropipettes was driven by a syringe pump at a speed of 0.1 μL min$^{-1}$. The animals were individual housed in an isolated room for 9 d. The animals were then anesthetized, and perfused with saline followed by 4% PFA. Brains were taken from the animals, and placed in 4% PFA overnight, and immersed in 30% sucrose in PBS. Slices were cut at 40-μm thickness using Leica VT 1000 vibratome (Leica Biosystems, German), and stained using DAPI, and imaged using FV 1000 (Olympus, Japan) or Nikon A1 confocal microscope (Nikon, Japan). The images were processed using Adobe Photoshop and Image J.

**TetLC expression**. Under the surgery conditions described above, glass micropipettes were used to inject AAV2-syn-EGFP-2A-TetLC-3flag (titer ~ 10$^{12}$ mL$^{-1}$, from Tai Tin Bio, Ltd., Shanghai, China) into the pyramid layer of the dorsal CA1 regions (the rat: AP, −3.8 mm, ML ± 2.8 mm, DV −2.6 mm), to express tetanus toxin light chain (TetLC), which is known to block the neurotransmitter release of the efferent projection terminals[11]. Infusion of the viruses (1 μL in CA1 per side) using glass micropipettes was driven by a syringe pump at a speed of 0.1 μL min$^{-1}$. After 30 days of the virus injection, the animals were subjected to behavioral studies.

After behavioral experiments, immunostaining and in situ hybridization were performed. Briefly, the mice were killed and perfused with PBS and PFA. After post-fixation in PFA at 4 °C overnight, the brains were sectioned with a cryostat (CM1900, Leica). Brain slices were incubated with rabbit anti-VAMP2 (1:300, Synaptic Systems) at 4 °C overnight, with biotinylated horse anti-rabbit antibody (1:500, Vector Labs) at room temperature for 3 h and then with Cy3-conjugated streptavidin (1:1000, Jackson ImmunoResearch) for 1 h. In situ hybridization of TetLC mRNA was performed, as described previously[54,55]. Images were captured with an epifluorescence microscope (Eclipse 80i, Nikon).

**Optogenetic implantation**. Both excitatory (AAV2-CaMKIIα-ChR2-EYFP) and inhibitory (AAV2-CaMKIIα-NpHR3.0-EYFP) optogenetic tools (titer ~ 10$^{12}$ mL$^{-1}$, from OBIO Technology, Ltd., Shanghai, China) were used to study the ipsCA1–conCA1 projections. Under the surgery conditions described above, guide cannulas for optic fibers and virus injections were implanted, and glass micropipettes via the guide cannulas injected the viruses into the pyramid layer of the

dorsal CA1 regions (the rat: AP, −3.8 mm, ML ± 2.8 mm, DV −2.6 mm). Injection of the viruses (1 μL in CA1 per side) was driven by a syringe pump at a speed of 0.1 μL min$^{-1}$. The guide cannulas for insertion of optic fibers (200 μm in diameter, from Biogene, Beijing, China) were affixed to the skull using dental cement with two stainless steel screws serving as anchors. Thirty days after the virus injection, the animals were subjected to behavioral studies. Before fear conditioning, the animals were allowed to adapt to the optic fibers inserted into the implanted guide cannulas for 3 d.

**Chemogenetic manipulation**. To excite the ipsCA1–conCA1 synapses for hours, and also minimizing the excitation of cell bodies, excitatory AAV2-hsyn-hM3Dq-mcitrine (titer ~ 10$^{12}$ mL$^{-1}$, from OBIO Technology, Ltd., Shanghai, China) was injected into the pyramid layer of ipsCA1 (1.5 μL; the rat: AP = −3.8 mm, ML = −2.8 mm, DV = −2.6 mm), while guide cannula was implanted into the stratum oriens of conCA1 (AP = −3.8 mm, ML = −2.8 mm, DV = −2.4 mm). Thirty days after the virus injection, the animals were subjected to behavioral studies. After fear conditioning, clozapine-N-oxide (CNO, 1.5 μg, 1.5 μL) was infused into conCA1 twice (immediately after fear conditioning and 30 min before retrieval tests at 8 h after fear conditioning) via the implanted guide cannula to excite persistently the ipsCA1 projection terminals onto conCA1.

Experimental procedures were used similar to previously described[56]. After 1 h of the behavioral studies, the rats were anesthetized with sodium pentobarbital (60 mg kg$^{-1}$) and perfused intracardially with PBS pH 7.4 followed by 4% paraformaldehyde (PFA). The isolated brains were post-fixed in 4% PFA overnight, and dehydrated in 30% sucrose in PBS at 4 °C. Then the brains were sectioned (40 μm thick coronal sections) by using a vibratome. C-Fos immunostaining was performed, as described previously. Free-floating sections were placed in a 0.01 M PBS solution containing 5% BSA and 0.3% Triton X-100 for 1 h followed by incubation with primary antibody: rabbit anti c-Fos (1:500, Santa Cruz) overnight at 4 °C, then washed slices for three times in PBS, followed by 2 h incubation with secondary antibody (Donkey anti rabbit Alexa-594, 1:1000 Life Technologies) at room temperature. Finally, the slices were washed three times, followed by mounting and cover-slipping on microscope slides. Images were acquired by using a confocal microscope (Nikon A1) with a 20 × objective at the same settings for all conditions.

**Electrophysiology**. Using previously described techniques[41–48,57], the field excitatory postsynaptic potentials (fEPSP) were recorded in brain slice (in mice) or anesthetized and freely moving rats. Brain slice: mouse brain was dissected and cut into slices at 400 μm thickness using Leica VT 1000 (Leica Biosystems), in ice-cold artificial cerebrospinal fluid (ACSF). Before fEPSP recordings, the slices were maintained for at least 30 min in the oxygenated (95% $O_2$ and 5% $CO_2$) warm (37 °C) ACSF containing (in mM) 120 NaCl, 2.5 KCl, 2 $CaCl_2$, 2 $MgSO_4$, 26 NaHCO$_3$, 1.25 NaH$_2$PO$_4$, 10 glucoses. Recording electrodes (4–6 MΩ) were pulled on a micropipette puller (Sutter Instruments, USA). Stimulating electrodes were made by gluing together a pair of twisted Teflon-coated 90% platinum/10% iridium wires (50-μm bare diameter, 100-μm coated diameter, World Precision Instruments, USA). Anesthetized and awake rats: with the anesthesia and surgery protocols described above, stimulating and recording electrodes were implanted to the targeted areas to find fEPSP. Both stimulating and recording electrodes were made by gluing together a pair of twisted Teflon-coated 90% platinum/10% iridium wires. For fEPSP study in freely moving rats, the implanted electrodes and stainless screws serving as ground and reference were fixed to the skull with dental cement. During recovery, the animals were handled and allowed to adapt to the recording environment in a soundproof chamber for 3 d before experiments.

**Fear conditioning**. The procedures for studying contextual fear memory and generalization were modified from those described previously[11–13,41,44,57,58]. Experiments were performed by using apparatuses from MED Associates Inc., Vermont (USA). The training box (T-box) and a non-training similar box (G-box) were differed in sizes, floors, walls, and lights, which were placed into the similar soundproof and ventilated chambers, and the behaviors were monitored by infrared video system. One day before fear conditioning, rats were allowed to acclimation to T-box and 1 h later to G-box by exploring each for 10 min. For slow generalization study, the animals were exposed to T-box but not G-box for 10 min on the acclimation day. On the day of fear conditioning, the rats were allowed free exploration for 2 min, and then received 5 footshocks (0.8 mA, 2 s duration) with averaged 2 min intervals, and returned to homecage 2 min after fear conditioning. During a single-foot fear conditioning, all the procedures were the same, excepting that the shock electrodes were banded by using adhesive tape directly to the front and back of the hind right leg ankle of the rats. Retrieval tests were performed by placing the rats into G-box first and 1 h later into T-box, both for 5 min, during which freezing levels were recorded for scoring generalization and fear memory through the computer system (MED Associates Inc.).

**Statistical analysis**. All values were reported as mean ± s.e.m. Except Fig. 3d (Student t-test) and Fig. 4d, f (repeated one-way ANOVA), repeated two-way or two-way ANOVA was used in the other figures, followed by parameter estimates

for control vs. treatment, and between treatments, or contrast effects for T-box vs. G-box. The significance level was set at $P < 0.05$.

**Data availability**. The authors declare that data supporting the findings of this study are available within the paper and its supplementary information files.

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

## Acknowledgements

Research was supported by 973 Program of the Ministry of Science and Technology of China (2013CB835100 and 2015CB553502 to L.X.), Strategic Priority Research Program of Chinese Academy of Sciences (XDB02020002 to L.X.) and External Cooperation Program BIC of Chinese Academy of Sciences (GJHZ1549 to G.R.-L.), Youth Innovation Promotion Association of Chinese Academy of Sciences (2013250 to Q.-X.Z.), National Science Foundation of China (31100775 and 31371141 to Q.-X.Z., 8157332 to Y.-Q.D., and U1502221 to L.X.), and Science and Technology Program of Yunnan Province (2013GA003 to L.X.). We thank Mr. Jin-Yang Liu and Ms. Jia-Lin Mi for assisting chemogenetic and slow generalization study, and Mr. Jin-Yang Liu and Ms. Ni-Ya Wang for assisting c-fos staining. We thank Dr. Mu-Ming Poo for suggesting the design of single-foot fear conditioning.

## Author contributions

H.Z. and G.-J.X. were responsible for the initial finding. H.Z., G.-J.X., L.J., N.-N.S. performed most of the experiments. D.-L.P. performed the study in mice. X.T. drew schematic figures. X.-B.H. and F.-Q.X. provided rabies virus and assisted tracing technique. J.-F.H., G.R.-L. L.-J.L., and R.-R.M. contributed to critical discussions and manuscript editing. Q.-X.Z., Y.-Q.D., and L.X. designed and directed the research and wrote the paper, which was reviewed by all authors.

## Additional information

**Competing interests:** The authors declare no competing financial interests.

