## [Peer Review File · Nature Communications]

Reviewers' comments:

Reviewer #1 (Remarks to the Author):

In this manuscript, Zhou et al. investigated the mechanisms underpinning contextual fear memory generalization. This is a very important and timely topic and the present work brings interesting and relevant findings that can potentially reshape our view on the subject.

Initially, they show that fear memory is rapidly formed, but generalization follows a distinct temporal pattern, being developed more gradually. Extinction memory also displays a distinct pattern regarding its retention (that is, suppression of fear) and its generalization (that is, renewal). The authors used a protocol where generalization develops rapidly (24h), and such differential temporal pattern suggests that in this case generalization did not occur due to the mere decay of memory` contextual details. It is then proposed that the phenomenon of generalization may be a unique active process of the brain that occurs separately from fear memory ones.

The authors hypothesize that generalization is mediated by interhemispheric information processing between the hippocampi. As an initial exploration of this possibility, unilateral or bilateral dorsal CA1 lesions were performed. Rats with bilateral lesion displayed amnesia, as expected. Interestingly, unilateral lesions prevented generalization (fear to a distinct context) while not affecting fear to the conditioned context. The same effect was found with pharmacological inactivation conducted before training and before retrieval. This specific set of data suggests that the development of generalization requires both hippocampi to take place.

In addition, they used a training procedure where animals were shocked in a single foot, and pharmacological inactivation of the ipsilateral or contralateral CA1 hemispheres was conducted. Pre-retrieval ipsCA1 and conCA1 inactivation had no effect on fear to the conditioned context, but both impaired generalization. It indicates that within the hippocampus laterality is not important for generalization, but both hippocampi are required for such outcome. They also looked into the effect of ipsi and contralateral inactivation of the posterior intralaminar nuclei of the thalamus, responsible for signaling foot shock information to the hippocampus via contralateral projections. Pre-training contralateral inactivation prevented the context+shock association and resulted in memory impairment. Ipsilateral inactivation, however, did not affect contextual fear memory formation but prevented generalization. The same procedure had no effect on retrieval regarding fear to the conditioned context but again impaired generalization. It is additional evidence that interhemispheric connectivity is underlying the development of contextual generalization.

These elegant set of experiments strongly indicate that bilateral CA1 cross-talk is essential for contextual fear generalization. With these intriguing finding in hands, the authors sought to determine the precise anatomical pathways responsible for it. For that, they used rabies virus to trace interhemispheric CA1 connections and optogenetic stimulation of these connections to check for fEPSP. This approach resulted in an amazing characterization of interhemispheric CA1 connectivity. The role of such connections upon generalization was then directly assessed using two approaches. By blocking the identified projections with both tetanus toxin light chain and optogenetics, the authors replicated their initial lesion findings of impaired generalization. Regarding optogenetic, this effect was observed both when CA1 was inactivated unilaterally and when projections from ipsCA1 to conCA1 were specifically silenced. Hence, the circuitry they carefully characterized appears to be indeed responsible for generalization.

Finally, changes in the ipsCA1-conCA1 fEPSP were recorded in freely moving rats in the hours following fear conditioning. In a nice parallel with the onset of generalization in their protocol,

potentiation was observed 24h after conditioning. Interestingly, the decreased on generalization following extinction 0.5 h post-training was reversed by activation of ipsCA1-conCA1 projections. That is, generalization can be accelerated or impaired by specific manipulations of these projections. These findings challenge the view of generalization as a mere consequence of a) the fade of memory contextual content due to forgetting or b) memory disengagement from HPC circuits and storage in neocortical areas. Instead, it proposes it results from active information processing in the hippocampus. Although the notion of the brain generalizing memory “on purpose” can at first glance be puzzling, the authors did a good job in contextualizing its importance - it makes memories more useful since only rarely previous events recur in an identical manner. The study looks very well planned and conducted. The authors used a variety of techniques to test their hypothesis. From lesions to the characterization of specific anatomical projections, and then precise manipulation of these projections, this work offers a systematical investigation of the neurobiological mechanisms underpinning memory generalization. Hence, I congratulate the authors for their amazing study.

However, I have some concerns that should be addressed.

Specific concerns:

- In the extinction experiments, freezing to the G-box are purely interpreted as an effect upon generalization, but there are alternative interpretations. For instance, in the 24h group, it is not possible to conclude that low freezing to the G-box is directly due to impaired generalization since fear to the T-box is also impaired. That is, low freezing levels at G-box probably occurs because the fear memory is inhibited, making any assumption about generalization hypothetical. Regarding the other groups (0.5h and 12h), the impaired generalization could be explained by the extended experience in the T-box, what may have allowed the encoding of more contextual details, thus increasing memory precision. The authors should acknowledge these possibilities.

- I have some concerns with some of the statistical analysis:

When performance is analyzed over sequential trials or blocks within a session (figures 1B, 1D, 1F, and 4B) data must be analyzed with repeated-measures ANOVA. When the performance of several groups (such as Sham, Uni and Bil in figurea 2B) are compared across distinct sessions (test in T-box vs G-box) a factorial ANOVA is expected to evaluate effects of group x session.

- Results section, line 119: The authors describe the first ips vs con experiment but an explanation of the single-foot fear conditioning procedure is given only at line 131. This should be reorganized to improve clarity. Or did the authors, in fact, ran a uni vs bil experiment there?

- Memory generalization has been reported in many different species, tasks, and protocols. Usually, it is observed at remote time points. In rodents, contextual fear generalization typically is reported following several weeks from training. That`s why systems consolidation (memory storage gradual reorganization from the hippocampus towards neocortical areas) have been pointed as a putative neurobiological mechanism for memory`s generalization. It is hypothesized that contextually rich memories would always require the hippocampus and when they shift to the cortex they become contextually poor, more schematic. In the contextual fear conditioning protocol of this work, generalization occurs much more rapidly (in a period of 24h). I don`t see it as a shortcoming because as the authors mentioned, generalization on some occasions (maybe most of them in our daily life) is all about recall flexibility. However, rapid generalization and gradual ones that correlate with systems consolidation probably follow different mechanisms. I encourage the authors to address this important issue. Given the potential impact of the current findings in the field, a careful discussion about generalization time course and possible distinct mechanisms is required.

Reviewer #2 (Remarks to the Author):

The manuscript of Zhou et al describes a series of experiments designed to differentiate the contribution of the dorsal CA1 region of the hippocampus to both context fear and fear generalization. Specifically the authors argue that an interhemispheric CA1 to CA1 transmission is required for generalization but not context fear learning. While these data are interesting, I feel the conclusions of the authors are not fully supported and have several serious reservations about the experiments, the statistics employed and the authors' interpretations.

Major points:

- 1) The authors repeatedly state their data is 'consistent with previous reports' when that is not the case. For example, my biggest concern with these data is their completely unprecedented timeline of context fear generalization. The papers they cite their consistency with, Wiltgen and Silva 2007 and Cullen et al 2015, detect the development of context fear generalization 2 to 4 weeks after training, whereas in the current study generalization is present at 24hrs. While the Cullen et al study uses very similar training parameters they find virtually no generalization at 24hrs (see their Figure 1a). This is clearly not consistent with the current data and calls into question if what is being studied here is the same process previously described in the literature.
- 2) The authors' interpretation of their data in different fear experiments as indicating 'disrupted generalization' versus an overall reduction in freezing behavior seems arbitrary and complicated with a poor use of statistics. For example in Figure 2b the authors claim that pretraining bilateral lesions 'impair both fear memory and generalization' however very similar freezing scores in figure 2g (ips. Group CNQX+TTX) is interpreted as impairing only generalization. Also in Figure 2h the authors concluded inhibition of the hippocampus with CNQX+TTX impairs only generalization when freezing in the conditioned context is reduced from the control group. A further example is in Figure 3n where the authors claim optogenetic inhibition of contralateral CA1 fibers impairs generalization but it looks to actually increase freezing in the conditioned context. The underlying problem is that the statistics sections of the methods is not clear on how analyses of these data was conducted. For most experiments there are 3 treatment groups (Control and either bi/uni or ips/con) and 2 contexts (G and T), thus the most appropriate statistical approach would be a 2-way ANOVA across group and context and post-hoc tests to look for differences between contexts within a given group. A more rigorous use of statistics would greatly simplify and standardize the interpretation of all the results.
- 3) In describing the muscimol experiments, both in the text on pg. 5 and in Figure 1d, the authors to ipsCA1 and conCA1, however it appears they are using a standard fear conditioning protocol here, thus these terms are meaningless- ipsi and contra to what?
- 4) The interpretation of the PIL experiments is crucially based on the fact that the inputs to PIL are contralateral and that PIL projects to hippocampus. However the reference the authors' provide is a study in humans. Data from the rat should be referenced.
- 5) In both the discussion and abstract the authors state 'Our data indicate that generalization is abstracted from bilaterally duplicated CA1 ensembles and encoded by gradual synaptic learning in ipsiCA1-conCA1 circuit'. This is a gross over interpretation of their results and not directly supported by the data presented. They offer absolutely no evidence of 'ensemble duplication'.
- 6) I do not understand the authors' design and interpretation of the Gq DREADD stimulation experiments (Figure 4). Why would non-specific stimulation of a large fraction of neurons in one hemisphere of CA1 specifically strengthen a memory trace? What is the justification for 8+ hours of

stimulation being necessary? Further, the authors provide no verification of CNO-mediated increase in activity (for example cFos induction).

7) The CA1-CA1 physiology in Figure 4 excludes necessary control groups. In Figure 4c there should be both a 0.5 hr and a 24hr control and in Figure 4f there should be a no extinction and home cage control.

8) The authors claim to restrict their viral injection to str. oriens or str. radiatum. I find this hard to believe given the volumes used and the example photos in the manuscript. Further finding contralateral labeling of CA1 pyramidal cells following injection of the RV-dG virus only argues that these neurons have synapses in the area of injection but does not demonstrate the post-synaptic target of these connections are other CA1 pyramidal cells.

9) At a minimum, anti-VAMP2 staining should be employed to verify efficacy of the TetLC expression and function.

10) Overall the methods are sparse and not well documented.

11) The first paragraph of the introduction, quoting Heraclitus and discussing disease, is completely unnecessary.

12) What were the stereotaxic coordinates used for the mouse experiments?

Point-by-point response to the referees' comments:

Reviewer #1 (Remarks to the Author):

In this manuscript, Zhou et al. investigated the mechanisms underpinning contextual fear memory generalization. This is a very important and timely topic and the present work brings interesting and relevant findings that can potentially reshape our view on the subject.

Initially, they show that fear memory is rapidly formed, but generalization follows a distinct temporal pattern, being developed more gradually. Extinction memory also displays a distinct pattern regarding its retention (that is, suppression of fear) and its generalization (that is, renewal). The authors used a protocol where generalization develops rapidly (24h), and such differential temporal pattern suggests that in this case generalization did not occur due to the mere decay of memory` contextual details. It is then proposed that the phenomenon of generalization may be a unique active process of the brain that occurs separately from fear memory ones.

The authors hypothesize that generalization is mediated by interhemispheric information processing between the hippocampi. As an initial exploration of this possibility, unilateral or bilateral dorsal CA1 lesions were performed. Rats with bilateral lesion displayed amnesia, as expected. Interestingly, unilateral lesions prevented generalization (fear to a distinct context) while not affecting fear to the conditioned context. The same effect was found with pharmacological inactivation conducted before training and before retrieval. This specific set of data suggests that the development of generalization requires both hippocampi to take place.

In addition, they used a training procedure where animals were shocked in a single foot, and pharmacological inactivation of the ipsilateral or contralateral CA1 hemispheres was conducted. Pre-retrieval ipsCA1 and conCA1 inactivation had no effect on fear to the conditioned context, but both impaired generalization. It indicates that within the hippocampus laterality is not important for generalization, but both hippocampi are required for such outcome. They also looked into the effect of ipsi and contralateral inactivation of the posterior intralaminar nuclei of the thalamus, responsible for signaling foot shock information to the hippocampus via contralateral projections. Pre-training contralateral inactivation prevented the context+shock association and resulted in memory impairment. Ipsilateral inactivation, however, did not affect contextual fear memory formation but prevented generalization. The same procedure had no effect on retrieval regarding fear to the conditioned context but again impaired generalization. It is additional evidence that interhemispheric connectivity is underlying the development of contextual generalization.

These elegant set of experiments strongly indicate that bilateral CA1 cross-talk is essential for contextual fear generalization. With these intriguing finding in hands, the authors sought to determine the precise anatomical pathways responsible for it. For that, they used rabies virus to trace interhemispheric CA1 connections and optogenetic stimulation of these connections to check for fEPSP. This approach resulted in an amazing characterization of interhemispheric CA1 connectivity. The role of such connections upon generalization was then directly assessed using two approaches. By blocking the identified projections with both tetanus toxin light chain

and optogenetics, the authors replicated their initial lesion findings of impaired generalization. Regarding optogenetic, this effect was observed both when CA1 was inactivated unilaterally and when projections from ipsCA1 to conCA1 were specifically silenced. Hence, the circuitry they carefully characterized appears to be indeed responsible for generalization.

Finally, changes in the ipsCA1-conCA1 fEPSP were recorded in freely moving rats in the hours following fear conditioning. In a nice parallel with the onset of generalization in their protocol, potentiation was observed 24h after conditioning. Interestingly, the decreased on generalization following extinction 0.5 h post-training was reversed by activation of ipsCA1-conCA1 projections. That is, generalization can be accelerated or impaired by specific manipulations of these projections.

These findings challenge the view of generalization as a mere consequence of a) the fade of memory contextual content due to forgetting or b) memory disengagement from HPC circuits and storage in neocortical areas. Instead, it proposes it results from active information processing in the hippocampus. Although the notion of the brain generalizing memory “on purpose” can at first glance be puzzling, the authors did a good job in contextualizing its importance - it makes memories more useful since only rarely previous events recur in an identical manner. The study looks very well planned and conducted. The authors used a variety of techniques to test their hypothesis. From lesions to the characterization of specific anatomical projections, and then precise manipulation of these projections, this work offers a systematical investigation of the neurobiological mechanisms underpinning memory generalization. Hence, I congratulate the authors for their amazing study.

Response: We would like to thank you very much for these wonderful comments.

However, I have some concerns that should be addressed.

Specific concerns:

- In the extinction experiments, freezing to the G-box are purely interpreted as an effect upon generalization, but there are alternative interpretations. For instance, in the 24h group, it is not possible to conclude that low freezing to the G-box is directly due to impaired generalization since fear to the T-box is also impaired. That is, low freezing levels at G-box probably occurs because the fear memory is inhibited, making any assumption about generalization hypothetical. Regarding the other groups (0.5h and 12h), the impaired generalization could be explained by the extended experience in the T-box, what may have allowed the encoding of more contextual details, thus increasing memory precision. The authors should acknowledge these possibilities.

Response: We fully agree with you and appreciate you very much for giving us these thoughtful knowledges.

- I have some concerns with some of the statistical analysis:

When performance is analyzed over sequential trials or blocks within a session (figures 1B, 1D, 1F, and 4B) data must be analyzed with repeated-measures ANOVA. When the performance of several groups (such as Sham, Uni and Bil in figure 2B) are compared across distinct sessions (test in T-box vs G-box) a factorial ANOVA is expected to evaluate effects of group x session.

Response: Thank you very much for these detailed and valuable suggestions, for which reviewer #2 also gave us the similar suggestions. Now, following exactly your and reviewer #2's suggestions, statistical comparisons are fully revised. Excepting that *t*-test was used in figure 3d, and repeated ANOVA was used in the figures 1b, 1e, 1f, 4d and 4f, the other statistical comparisons in main figures and extended data figures 2 and 3 were all changed into using two-way ANOVA.

All statistical data can be found in the supplementary information tables, and these were also described in Statistical analyses of Methods, and figure legends of revised manuscript.

- Results section, line 119: The authors describe the first ips vs con experiment but an explanation of the single-foot fear conditioning procedure is given only at line 131. This should be reorganized to improve clarity. Or did the authors, in fact, ran a uni vs bil experiment there?

Response: We apologize for making this confusion, for which reviewer #2 also gave the similar comment. Now, we clarified this term as left CA1 vs right CA1 in the figure and text, because this experiment was specifically designed to test the possible laterality on generalisation that was dependent on bilateral CA1 regions.

- Memory generalization has been reported in many different species, tasks, and protocols. Usually, it is observed at remote time points. In rodents, contextual fear generalization typically is reported following several weeks from training. That's why systems consolidation (memory storage gradual reorganization from the hippocampus towards neocortical areas) have been pointed as a putative neurobiological mechanism for memory's generalization. It is hypothesized that contextually rich memories would always require the hippocampus and when they shift to the cortex they become contextually poor, more schematic. In the contextual fear conditioning protocol of this work, generalization occurs much more rapidly (in a period of 24h). I don't see it as a shortcoming because as the authors mentioned, generalization on some occasions (maybe most of them in our daily life) is all about recall flexibility. However, rapid generalization and gradual ones that correlate with systems consolidation probably follow different mechanisms. I encourage the authors to address this important issue. Given the potential impact of the current findings in the field, a careful discussion about generalization time course and possible distinct mechanisms is required.

Response: Your suggestion is excellent for us to break through our fossilized thinking frame about generalisation. Following your suggestion, we added a short discussion for rapid (hours) and gradual (weeks) generalization, for which possible distinct mechanisms might be required, in Discussion section: "Considering previous findings, we would like to propose that the formation of generalisation is gradual but have actively rapid and passively slow phases, likely corresponding to cellular consolidation and systems consolidation of memory when the contextual components are more or less specific." in lines 13 pages 11 of the revised manuscript.

Reviewer #2 (Remarks to the Author):

The manuscript of Zhou et al describes a series of experiments designed to differentiate the contribution of the dorsal CA1 region of the hippocampus to both context fear and fear generalization. Specifically the authors argue that an

interhemispheric CA1 to CA1 transmission is required for generalization but not context fear learning. While these data are interesting, I feel the conclusions of the authors are not fully supported and have several serious reservations about the experiments, the statistics employed and the authors' interpretations.

Response: We appreciate you very much for your positive comments and suggestions that substantially improved our manuscript.

Major points:

1) The authors repeatedly state their data is 'consistent with previous reports' when that is not the case. For example, my biggest concern with these data is their completely unprecedented timeline of context fear generalization. The papers they cite their consistency with, Wiltgen and Silva 2007 and Cullen et al 2015, detect the development of context fear generalization 2 to 4 weeks after training, whereas in the current study generalization is present at 24hrs. While the Cullen et al study uses very similar training parameters they find virtually no generalization at 24hrs (see their Figure 1a). This is clearly not consistent with the current data and calls into question if what is being studied here is the same process previously described in the literature.

Response: We appreciate you very much for pointing out this question. The statement was changed into "This rapid generalisation is differed from the more gradual one reported previously showing that several weeks are required for generalisation development in mice" in the Results (lines 10 pages 4). Following the suggestions from you and reviewer #1, we added a short discussion for rapid (hours) and gradual (weeks) generalization, for which possible distinct mechanisms might be required, in Discussion section: "Considering previous findings, we would like to propose that the formation of generalisation is gradual but have actively rapid and passively slow phases, likely corresponding to cellular consolidation and systems consolidation of memory when the contextual components are more or less specific." in lines 13 pages 11 of the revised manuscript.

2) The authors' interpretation of their data in different fear experiments as indicating 'disrupted generalization' versus an overall reduction in freezing behavior seems arbitrary and complicated with a poor use of statistics. For example in Figure 2b the authors claim that pretraining bilateral lesions 'impair both fear memory and generalization' however very similar freezing scores in figure 2g (ips. Group CNQX+TTX) is interpreted as impairing only generalization. Also in Figure 2h the authors concluded inhibition of the hippocampus with CNQX+TTX impairs only generalization when freezing in the conditioned context is reduced from the control group. A further example is in Figure 3n where the authors claim optogenetic inhibition of contralateral CA1 fibers impairs generalization but it looks to actually increase freezing in the conditioned context. The underlying problem is that the statistics sections of the methods is not clear on how analyses of these data was conducted. For most experiments there are 3 treatment groups (Control and either bi/uni or ips/con) and 2 contexts (G and T), thus the most appropriate statistical approach would be a 2-way ANOVA across group and context and post-hoc tests to look for differences between contexts within a given group. A more rigorous use of statistics would greatly simplify and standardize the interpretation of all the results.

Response: Thank you very much for these detailed and valuable suggestions, for which reviewer #1 also gave us the similar suggestions. Now, following exactly your

and reviewer #1's suggestions, statistical comparisons are fully revised. Excepting that *t*-test was used in figure 3d, and repeated ANOVA was used in the figures 1b, 1e, 1f, 4d and 4f, the other statistical comparisons in main figures and extended data figures 2 and 3 were all changed into using two-way ANOVA.

All statistical data can be found in the supplementary information tables, and these were also described in Statistical analyses of Methods, and figure legends of revised manuscript.

3) In describing the muscimol experiments, both in the text on pg. 5 and in Figure 1d, the authors to ipsCA1 and conCA1, however it appears they are using a standard fear conditioning protocol here, thus these terms are meaningless- ipsi and contra to what?

Response: We apologize for making this confusion, for which reviewer #1 also gave the similar comment. Now, we clarified this term as left CA1 vs right CA1 in the figure and text, because this experiment was specifically designed to test the possible laterality on generalisation that was dependent on bilateral CA1 regions.

4) The interpretation of the PIL experiments is crucially based on the fact that the inputs to PIL are contralateral and that PIL projects to hippocampus. However the reference the authors' provide is a study in humans. Data from the rat should be referenced.

Response: We apologize for this error. The reference "21. Pakhomova, A.S. Diencephalic afferents of the rat hippocampus. *Neurophysiol.* **13**, 359-364 (1981)." from the rat study was cited.

5) In both the discussion and abstract the authors state 'Our data indicate that generalization is abstracted from bilaterally duplicated CA1 ensembles and encoded by gradual synaptic learning in ipsiCA1-conCA1 circuit'. This is a gross over interpretation of their results and not directly supported by the data presented. They offer absolutely no evidence of 'ensemble duplication'.

Response: We fully agree with you. Considering also the suggestion from reviewer #1, we revised the sentence in the abstract as "Thus, rapid generalisation is an active process depended on bilateral CA1 regions, and encoded by gradual synaptic learning in ipsCA1-conCA1 circuit". Also, "CA1 ensembles" in the Discussion section was changed into "CA1 regions", in lines 23 pages 11 of revised manuscript.

6) I do not understand the authors' design and interpretation of the Gq DREADD stimulation experiments (Figure 4). Why would non-specific stimulation of a large fraction of neurons in one hemisphere of CA1 specifically strengthen a memory trace? What is the justification for 8+ hours of stimulation being necessary? Further, the authors provide no verification of CNO-mediated increase in activity (for example cFos induction).

Response: You asked excellent questions for which we do not fully understand yet. Because we found that rapid generalization is a gradual form of learning in contrast to fear memory itself, we assumed that the ipsCA1-conCA1 synapses must be active for a certain period of time. This is the reason why Gq DREADD rather optogenetic

ChR2 was used, for which CNO was given twice, immediately after fear conditioning and 30 min before retrieval tests at 8 h.

We fully agree with you for how this ‘artificial’ enhancement of the ipsCA1-conCA1 synaptic efficacy by Gq DREADD could strengthen a memory trace. This experiment was performed after all of the experiments were finished because we wanted further to know how important it is. Nevertheless, we do not know how to interpret this result in detail, but still would like to speculate. Fear memory is somehow duplicated in bilateral CA1 regions to endow generalisation, and the recurrent circuit through ipsCA1-conCA1 synapses may build an additional access to these memory resources, therefore strengthening the memory trace. At the same time, this ‘artificially’ enhanced recurrent circuit may speed up the active process for abstracting the common elements from the original memories following the principle of ‘firing together wiring together’. This speculation may be an interesting direction in the future study.

Also, following your suggestion, c-fos staining after the treatment of CNO or saline was performed and shown in extended data figure 10 d-g, in which c-fos expression was enhanced by CNO relative to saline treatment.

7) The CA1-CA1 physiology in Figure 4 excludes necessary control groups. In Figure 4c there should be both a 0.5 hr and a 24hr control and in Figure 4f there should be a no extinction and home cage control.

Response: By following exactly your suggestions, these control experiments are performed, and shown in Figure 4c, 4d and 4f of the revised manuscript.

8) The authors claim to restrict their viral injection to str. oriens or str. radiatum. I find this hard to believe given the volumes used and the example photos in the manuscript. Further finding contralateral labeling of CA1 pyramidal cells following injection of the RV-dG virus only argues that these neurons have synapses in the area of injection but does not demonstrate the post-synaptic target of these connections are other CA1 pyramidal cells.

Response: We fully agree with you the comment, and thus weakened our conclusion about the result “it is possible that ipsCA1 receives bilateral CA3 inputs mainly at stratum radiatum but it can sense conCA1 inputs mainly at stratum oriens.” in lines 21 pages 7 of revised manuscript. Since the dense pyramidal neurons in CA1 may be like a separation barrier between str. oriens and str. radiatum, the injection of AAV at str.

oriens or str. radiatum with the injection volume (1 μ L) at the injection speed of 0.1 μ L/min may still keep AAV not easily spreading from one to the other.

[Redacted]

[Redacted]

[Redacted]

9) At a minimum, anti-VAMP2 staining should be employed to verify efficacy of the TetLC expression and function.

Response: Following your suggestion, we performed anti-VAMP2 staining in which VAMP2 was obviously decreased in conCA1, while TetLC mRNA was expressed in ipsCA1 where the virus was injected. This result is showed as extended data figure 9.

10) Overall the methods are sparse and not well documented.

Response: Methods are thoroughly revised, and more references are cited.

11) The first paragraph of the introduction, quoting Heraclitus and discussing disease, is completely unnecessary.

Response: We removed these sentences according to your suggestion.

12) What were the stereotaxic coordinates used for the mouse experiments?

Response: In the mouse experiments (showing only in extended data figures), the stereotaxic coordinates for the CA1 stratum oriens are AP = -2.0 mm, ML = - 1.3 mm, DV = - 1.25 mm, and those for the CA1 stratum radiatum are the same but DV = - 1.35 mm; those for the midline of the dorsal hippocampal commissure (DHC) below corpus callosum are AP = -0.9 mm, ML = 0 mm, DV= - 2.0 mm. These are described in Viral tracing of Methods of revised manuscript.

Finally, we would like to sincerely thank both the reviewers for their substantial contributions in improving our manuscript, and also for giving us so many excellent questions and ideas for the future study.

Reviewers' comments:

Reviewer #1 (Remarks to the Author):

beautiful work by the authors.

They have addressed all the issues previously raised in the first submission,

Reviewer #2 (Remarks to the Author):

In the revised Zhuo et al manuscript the authors do address some of the concerns raised by the reviewers, however I remain unswayed regarding the validity of their main conclusions based on a poor explanation of the key underlying behavior.

1. As I raised in my initial comments, the authors' behavioral data, even under control conditions, is at odds with the existing literature, including the papers they cite. Simply distinguishing the behavior they report as "rapid generalization" does not alter the fact that it is completely unprecedented to see equal freezing in a distinct untrained context 24hrs following fear conditioning using this protocol. I have reexamined the literature myself searching for a plausible explanation, however I have failed to find any similar results in the generalization literature. It is worth noting that in the context of the memory allocation hypothesis put forward by Cai et al in a recent paper (Cai DJ et al, Nature 2016) the current result could be attributable to the pre-exposure phase of the protocol, however this is not addressed conceptually or experimentally by the authors.

2. I thank the authors for including a more detailed report of their statistical analyses, however this does not solve the problem I raised considering selective interpretation. When is low freezing in both contexts just poor learning and when is it a lack of generalization? Moreover a significant difference in freezing levels in the G-box and T-box can be repeatedly found even under control conditions (Fig 2g, Fig 3j).

3. I am unclear why the authors employed the RVdG virus in these experiments. In the methods they reference a 2015 Wei et al paper, but those authors used the rabies for transsynaptic tracing by providing the G protein in trans. In the figure included in the reviewer responses it appears the starter cells are CA2 pyramidal cells. This needs to be clarified.

4. I still remain puzzled by the rationale and interpretation of the DREADD experiment. Frankly the new data in extended figure 10 did little to help clarify what is going on here. I realize the authors are applying the CNO locally to hemisphere contralateral to the site of virus injection, which they report leads to an increase in cFos expression in the postsynaptic CA1 pyramidal cells, but not the presynaptic neurons. Why should this be the case using this system?

Reviewer #3 (Remarks to the Author):

In contextual fear conditioning experiments, rodents will initially exhibit conditioned fear only to the trained context. However, over time, generalized freezing to non-trained contexts emerges. This generalization phenomenon is of interest since it likely improves memory-guided decision making by uncoupling a particular decision/response from the exact set of circumstances that were initially

encountered. From this perspective, I found this paper to be especially interesting. Zhou et al present a comprehensive set of experiments—including optogenetic, chemogenetic, viral tracing, behavioral analyses—that all converge on a single conclusion: That an inter-hemispheric CA1-CA1 pathway mediates the time-dependent emergence of generalization. I am enthusiastic about this paper—I think it is of broad interest, and has important implications (e.g., knowledge generation) and will provide the impetus for new lines of research.

I have a few suggestions that might improve presentation.

1. Time course of generalization. The authors acknowledge that the generalization they observe is much more rapidly developing than has been traditionally observed. They address this discrepancy in the discussion but only superficially. They should add more here, perhaps contrasting the putative mechanisms for slowly-developing vs. rapidly-developing generalization (e.g., Xu and Sudhof). They also refer to cellular vs. systems consolidation, and suggest that these might map onto rapid and slow generalization, but how?
2. Density. This is a dense paper with an impressive range and number of experiments. Presentation would be improved by breaking the paper down into more easily digestible chunks (i.e., paragraphs).
3. Clarity. For the most part I could follow the authors' arguments. But in places, there were gaps in logic, or sentences that were unclear. Some examples include, but are not limited to: lines 55-58 (doesn't flow from previous, since previous is suggesting something which is presumably adaptive); lines 145-147 (unclear).
4. Statistical analyses. In many instances the authors design experiments in which an interaction is predicted (e.g., contrasting effects of unilateral vs. bilateral CA1 manipulations on fear learning/generalization). However, in their ANOVAs they have not reported whether interaction terms are significant (which would then permit the direct contrasts that they go onto to test). Again, an example would be Fig. 2g-h, but there are multiple examples throughout the paper. The authors need to be more explicit about what kind of statistical support they have for the differential effects.
5. Figure 1. The second part of this figure was a little hard to follow. First, the motivation for the extinction experiments was unclear to me. I feel they could be omitted, and not detract from the main thrust of the paper (in fact this would improve clarity). If they are important, then there is a second point of confusion for me: At the time point used in Fig. 1g, do we know whether there is usually generalization? (The authors showed generalization at 24 hours following training but not beyond). That is, in the absence of the extinction training phase, is there generalization at this time point?

Point-by-point response to the referees' comments:

Reviewer #1 (Remarks to the Author):

Beautiful work by the authors.

They have addressed all the issues previously raised in the first submission.

Response: We appreciate very much for your substantial contributions to our manuscript.

Reviewer #2 (Remarks to the Author):

In the revised Zhuo et al manuscript the authors do address some of the concerns raised by the reviewers, however I remain unswayed regarding the validity of their main conclusions based on a poor explanation of the key underlying behavior.

Response: We appreciate very much for your time and suggestions, while feeling sorry for not catching on your ideas accurately. This is largely due to the difficulties how to propose a hypothesis for explaining the novel rapid generalisation while theoretically has to fitting the published literatures. In this revision of our manuscript, we hoped that we had done better in following your suggestions.

1. As I raised in my initial comments, the authors' behavioral data, even under control conditions, is at odds with the existing literature, including the papers they cite. Simply distinguishing the behavior they report as "rapid generalization" does not alter the fact that it is completely unprecedented to see equal freezing in a distinct untrained context 24hrs following fear conditioning using this protocol. I have reexamined the literature myself searching for a plausible explanation, however I have failed to find any similar results in the generalization literature. It is worth noting that in the context of the memory allocation hypothesis put forward by Cai et al in a recent paper (Cai DJ et al, Nature 2016) the current result could be attributable to the pre-exposure phase of the protocol, however this is not addressed conceptually or experimentally by the authors.

Response: We fully accept your criticism, and opened our mind to seek the possible answer.

First, we found that Land, C. & Riccio, D.C. (1999) reported in rat study that 24 h after one-trial inhibitory avoidance training, fear memory was impaired by a change in context, but not if pretraining treated with the NMDA receptor modulator D-cycloserine, implicating that this "rapid generalisation" to untrained context may indeed have occurred within 24 h under the drug-enhanced neuronal excitability, likely also related to the hippocampal functions.

Second, we appreciate your detailed suggestions, which are brilliant to explain how "rapid generalisation" could occur. Silva A.J. (2017) and others including Cai D.J. et al (Nature, 2016) have developed a memory hypothesis termed "allocate to link" (see Silva, A.J. Memory's Intricate Web. Sci Am. 317, 30-37, 2017. I translated this article

into Chinese by the request from a Chinese Press for popularization of important scientific findings). Before fear conditioning training, the same group of the animals were exposed to T-box and 1 h later to G-box so that the animals might have formed two memories for both the contexts, for which certain hippocampal neurons were chosen for storing the memories, possibly through activation of the NMDA receptors and synaptic plasticity. The beauty of the science here is that the two memories for the two contexts could be linked together, recalling one leading to recall of the other, according to the “allocate to link” hypothesis. Thus, it is possible that G-box will somehow further link to fear when T-box was associated with fear.

Third, it is a major finding of our present study that the ipsCA1 and conCA1 are linked gradually by increasing the ipsCA1-conCA1 synaptic weight for “rapid generalisation”.

Altogether, we added a discussion paragraph (last line, page 12) to present a possible explanation for “rapid generalisation”: A number of studies have demonstrated gradual generalisation, for which 14 d²³⁻²⁶ or even longer^{12, 13, 27, 28} are required for its formation. Theories have been developed for understanding this time-dependent process that may relate to systems consolidation of memory when the contextual components become less specific^{13, 27}. In contrast, early generalisation within 2 d after fear conditioning training can occur by the learning process in a neural circuit that involves the mPFC, the thalamus, and the hippocampus¹¹. Furthermore, early or rapid generalisation may have occurred within 1 d if the rats were treated with D-cycloserine, an enhancer of NMDA receptor activity, before one-trial inhibitory avoidance training²⁶. This is somewhat consistent with our present finding in which generalisation was developed rapidly within 24 h, likely due to pre-exposure to T-box and 1 h later to G-box by which NMDA receptors would have been activated before fear conditioning. Moreover, the context memories for T-box and G-box could be then linked together as a result of the temporally closed neuronal ensembles, as suggested by the “allocate to link” hypothesis^{29, 30}. All together, we would like to propose a hypothesis for which the formation of generalisation is gradual but have actively rapid and passively slow phases, likely corresponding to cellular consolidation and systems consolidation of memory when the contextual components are more or less specific.

Also, we added a sentence (last line, page 13) for explaining how the ipsCA1-conCA1 could be linked: Their activity history is likely to explain how bilateral CA1 neurons are selected for developing generalisation as suggested recently^{29, 30}.

2. I thank the authors for including a more detailed report of their statistical analyses, however this does not solve the problem I raised considering selective interpretation. When is low freezing in both contexts just poor learning and when is it a lack of generalization? Moreover a significant difference in freezing levels in the G-box and T-box can be repeatedly found even under control conditions (Fig 2g, Fig 3j).

Response: We now realized that our statement in figure 1b is not exactly accurate: “..., when recall efficiency became equivalent to that in T-box”. We revised “equivalent” into “near equivalent” (line 8, page 4). Also, we removed only from the phrase “impaired only generalisation” throughout the manuscript.

As to the questions “When is low freezing in both contexts just poor learning and when is it a lack of generalization?”, our understanding is that learning is the acquisition and storage processing of the information while memory is the stored information dependent on learning and later retrieval. In our study, the manipulations before or after training resulted in low freezing in both contexts, and this might attribute to poor learning (Fig. 2b, g, Fig. 3f, and extended data Fig. 2b, f; Bi or con panel). The other manipulations such as before or during retrieval led to normal or slightly reduced freezing in T-box were possibly due to a lack of generalization and/or poor learning as well, because we believe that “rapid generalisation” requires “slow internal learning”.

To the question “Moreover a significant difference in freezing levels in the G-box and T-box can be repeatedly found even under control conditions (Fig 2g, Fig 3j).”, we are not fully understand this phenomenon yet at the present stage. One possible explanation is that fear memory needs stability while generalisation demands more flexibility during recall so that rapid generalisation is often slightly lower than the original memory with statistical significance, even sometimes under different control conditions.

3. I am unclear why the authors employed the RVdG virus in these experiments. In the methods they reference a 2015 Wei et al paper, but those authors used the rabies for transsynaptic tracing by providing the G protein in trans. In the figure included in the reviewer responses it appears the starter cells are CA2 pyramidal cells. This needs to be clarified.

Response: We cited Wei et al in the Methods for the tracing techniques. In fact, we had first tried the injection of transsynaptic RV into unilateral dorsal CA1, but it resulted in labelling in multiple hippocampal areas bilaterally, which cannot ruled out the multiple-synaptic or indirect labelling. In order to achieve monosynaptic labelling, we used RVdG that actually labelled contralateral CA1 clearly with very few labels in other areas using the similar protocol.

[Redacted]

[Redacted]

[Redacted]

[Redacted]

[Redacted]

[Redacted]

4. I still remain puzzled by the rationale and interpretation of the DREADD experiment. Frankly the new data in extended figure 10 did little to help clarify what is going on here. I realize the authors are applying the CNO locally to hemisphere contralateral to the site of virus injection, which they report leads to an increase in cFos expression in the postsynaptic CA1 pyramidal cells, but not the presynaptic neurons. Why should this be the case using this system?

Response: Our finding at the first look may be unbelievable, as you asked in the previous review “Why would non-specific stimulation of a large fraction of neurons in one hemisphere of CA1 specifically strengthen a memory trace?.” However, following your suggestion about the “allocate to link” hypothesis, we had clearer idea to explain this mazing result of the DREADD experiment.

First, with the CNO locally injected into hemisphere contralateral to the side of virus injection to excite the terminals with Gq expression, it is reasonable that cFos expression was increased in the postsynaptic CA1 pyramidal cells, but not the presynaptic neurons, because, as our understanding, backpropagation of the excitation by the CNO from the terminals to cell body should be not usually possible under normal conditions due to the increasing axonal distribution of the potassium channels closing to cell body.

Second, our finding is still somewhat consistent with the “allocate to link” hypothesis proposed by Silva A.J. et al (2017). A large fraction of the excited postsynaptic neurons by the CNO would be at least mainly those that had received the contralateral projections, but not the others without such the CA1-CA1 monosynaptic connections. Thus, the postsynaptic neurons would be chosen in this case mainly based on the contralateral inputs. Nevertheless, this may be not sufficient for rapid generalisation, because the fear memory presumably allocated in bilateral CA1 is still not linked together for developing rapid generalisation, even though the ipsCA1-conCA1 synapses had nonselective stronger synaptic activities with the CNO treatment.

Third, the neuronal ensembles for the fear memory would possibly replay at times, presumably reflected at least partly by synchronized activity between ipsCA1 and conCA1, a higher possibility in many of these neurons allocated context memories as described previously, followed by the rule of “firing together wiring together” through spike-timing dependent plasticity mechanisms. This endogenous activity of the bilateral CA1 ensembles would then overlay on those synapses and postsynaptic neurons with the CNO-enhanced synaptic activity. Still, the core mechanism is the gradual developing CA1-CA1 synaptic potentiation, while the CNO speeds up the process so that generalisation may occur even earlier.

Therefore, we added a sentence to suggest the possible explanation (line 17, page 11) “To excite these synapses continuously for hours that might overlay on the endogenous active process of the selected neurons in developing a synaptic potentiation gradually,”

Overall, we would like to thank you very much for your excellent suggestions, which raised many important questions worth of further study in the future.

Reviewer #3 (Remarks to the Author):

In contextual fear conditioning experiments, rodents will initially exhibit conditioned fear only to the trained context. However, over time, generalized freezing to non-trained contexts emerges. This generalization phenomenon is of interest since it likely improves memory-guided decision making by uncoupling a particular decision/response from the exact set of circumstances that were initially encountered. From this perspective, I found this paper to be especially interesting. Zhou et al present a comprehensive set of experiments—including optogenetic, chemogenetic, viral tracing, behavioral analyses—that all converge on a single conclusion: That an inter-hemispheric CA1-CA1 pathway mediates the time-dependent emergence of generalization. I am enthusiastic about this paper—I think it is of broad interest, and has important implications (e.g., knowledge generation) and will provide the impetus for new lines of research.

Response: We appreciate your comments very much.

I have a few suggestions that might improve presentation.

1. Time course of generalization. The authors acknowledge that the generalization they observe is much more rapidly developing than has been traditionally observed. They address this discrepancy in the discussion but only superficially. They should add more here, perhaps contrasting the putative mechanisms for slowly-developing vs. rapidly-developing generalization (e.g., Xu and Sudhof). They also refer to cellular vs. systems consolidation, and suggest that these might map onto rapid and slow generalization, but how?

Response: We fully accept your constructive suggestions. Now, more detailed discussion was added in the revised manuscript (last line, page 12) to present a possible explanation for “rapid generalisation”: A number of studies have demonstrated gradual generalisation, for which 14 d²³⁻²⁶ or even longer^{12, 13, 27, 28} are required for its formation. Theories have been developed for understanding this time-dependent process that may relate to systems consolidation of memory when the contextual components become less specific^{13, 27}. In contrast, early generalisation within 2 d after fear conditioning training can occur by the learning process in a neural circuit that involves the mPFC, the thalamus, and the hippocampus¹¹. Furthermore, early or rapid generalisation may have occurred within 1 d if the rats were treated with D-cycloserine, an enhancer of NMDA receptor activity, before one-trial inhibitory avoidance training²⁶. This is somewhat consistent with our present finding in which generalisation was developed rapidly within 24 h, likely due to pre-exposure to T-box and 1 h later to G-box by which NMDA receptors would have been activated before fear conditioning. Moreover, the context memories for T-box and G-box could be then linked together as a result of the temporally closed neuronal ensembles, as suggested by the “allocate to link” hypothesis^{29, 30}. All together, we would like to propose a hypothesis for which the formation of generalisation is gradual but have actively rapid and passively slow phases, likely corresponding to cellular consolidation and systems consolidation of memory when the contextual components are more or less specific.

2. Density. This is a dense paper with an impressive range and number of experiments. Presentation would be improved by breaking the paper down into more easily digestible chunks (i.e., paragraphs).

Response: More paragraphs were now separated by independent question or experiment, following your suggestions.

3. Clarity. For the most part I could follow the authors’ arguments. But in places, there were gaps in logic, or sentences that were unclear. Some examples include, but are not limited to: lines 55-58 (doesn’t flow from previous, since previous is suggesting something which is presumably adaptive); lines 145-147 (unclear).

Response: We apologize for our unclear writing.

The sentences in lines 55-58 were revised as following (indicated by red colour): “..., a time-dependent capacity developed following the original memory. However, it is not fully understood how this capacity is evolved from the original memories. This is also a primary question in artificial intelligence research fields regarding how to

endow better generalisation accuracy in dealing unpredicted **but similar** circumstances that the “neural network” has never been trained. **In addition**, overgeneralisation of fear memories is one of the key symptoms in posttraumatic stress disorder (PTSD), which may occur with a passage of time after trauma. **Thus**, elucidating the neural basis of generalisation holds the potential impact on these fields.”

The sentences in lines 145-147 were revised as following (indicated by red colour): “Thus, these findings indicate that **such asymmetry in the PIL, conPIL responsible for both fear memory and generalisation while ipsPIL responsible for generalisation, is somehow emerged into a symmetry in which bilateral CA1 is responsible for generalisation while unilateral CA1 is sufficient for fear memory.**”

Other sentences if we felt not clear were also revised (indicated in red colour).

4. Statistical analyses. In many instances the authors design experiments in which an interaction is predicted (e.g., contrasting effects of unilateral vs. bilateral CA1 manipulations on fear learning/generalization). However, in their ANOVAs they have not reported whether interaction terms are significant (which would then permit the direct contrasts that they go onto to test). Again, an example would be Fig. 2g-h, but there are multiple examples throughout the paper. The authors need to be more explicit about what kind of statistical support they have for the differential effects.

Response: We apologize for not explicitly pointing out the details of the statistical analyses. Now, all the details in the Results and Figure legend were added, which were indicated as red colour throughout the paper.

5. Figure 1. The second part of this figure was a little hard to follow. First, the motivation for the extinction experiments was unclear to me. I feel they could be omitted, and not detract from the main thrust of the paper (in fact this would improve clarity). If they are important, then there is a second point of confusion for me: At the time point use in Fig. 1g, do we know whether there is usually generalization? (The authors showed generalization at 24 hours following training but not beyond). That is, in the absence of the extinction training phase, is there generalization at this time point?

Response: We apologize for unclear writing. Now, the second part of the figure 1 was fully revised as following (indicated by red colour): “**To explore the characteristics of this novel generalisation in more details, which might be useful for studying its underlying mechanism**, we measured the generalisation and fear memory 24 h after extinction training that was performed by exposing the rats into T-box for 50 min (Fig. 1c), starting at 0.5, 12 or 24 h after fear conditioning for the separate groups (Fig. 1d, Δh). **These time points represented increasing levels (low to high) of the generalisation that was then maintained for at least 7 d (see Veh groups in Extended data Fig. 2e, f).** Consistent with previous report¹⁴, extinction training at the early stages (0.5 and 12 h) had nonsignificant effect on fear memory, but that at the later time point (24 h) largely reduced fear memory (i.e. suppression) (Fig. 1e, T-box: **0.5 h vs. 24 h, ### $P < 0.001$, and # $P < 0.03$, 12 h vs. 24 h; parameter estimates, repeated two-way ANOVA).** In contrast, extinction training at all these stages (0.5, 12, and 24 h) disrupted its generalisation (i.e. renewal) (Fig. 1e, G-box: **T-box vs. G-box at the time points, *contrast effects, repeated two-way ANOVA).** This revealed also a distinct pattern of generalisation in memory extinction. Retraining after extinction

caused gradual generalisation reformation over 24 h again (Fig.1f, T-box vs. G-box at Δh , *contrast effects, repeated two-way ANOVA). ”

We would like to express our sincere thanks to the dear three reviewers who would have spent a lot of their precious time to review our paper, and to give us these wonderful comments, pertinent criticisms, constructive suggestions. We ourselves have really felt that our manuscript is substantially improved following their suggestions.

Also, from these suggestions, we have been learning many knowledges that would have “generalized” into new research questions or directions.

Reviewers' comments:

Reviewer #2 (Remarks to the Author):

The authors have addressed my comments on the previous version of the manuscript and I find the paper much improved. While I still remain puzzled how synaptic strengthening between left and right CA1 pyramidal cells, which presumably encode the conditioned context as "place cells", via a hitherto unknown circuit leads to contextual fear generalization, I am hopefully this study will stimulate future studies to reexamine these processes.

Reviewer #3 (Remarks to the Author):

I am not satisfied by a number of the responses to my comments, as well as those by reviewer 2. I will focus my comments below to the major weaknesses remaining in the paper.

1. Explanation of rapid generalization. I am afraid that is still unclear why these authors observe rapid generalization, whereas the vast majority of other studies do not. The authors hint that the behavioral protocol that they use might be responsible, but the only way to address this would be to for the authors to show that you get slow generalization when using more standard training protocols. Without this contrast, the discussion provided on p. 12 remains quite speculative (and also not especially clear).

2. Statistical analyses. I requested that they provide more statistical details. In particular, for some of their claims they would require significant interaction terms in their ANOVAs. These have not been provided.

Point-by-point response to the referees' comments:

Reviewer #2 (Remarks to the Author):

The authors have addressed my comments on the previous version of the manuscript and I find the paper much improved. While I still remain puzzled how synaptic strengthening between left and right CA1 pyramidal cells, which presumably encode the conditioned context as “place cells”, via a hitherto unknown circuit leads to contextual fear generalization, I am hopefully this study will stimulate future studies to reexamine these processes.

Response: We appreciate you very much for your substantial contributions to our manuscript.

Reviewer #3 (Remarks to the Author):

I am not satisfied by a number of the responses to my comments, as well as those by reviewer 2. I will focus my comments below to the major weaknesses remaining in the paper.

1. Explanation of rapid generalization. I am afraid that is still unclear why these authors observe rapid generalization, whereas the vast majority of other studies do not. The authors hint that the behavioral protocol that they use might be responsible, but the only way to address this would be to for the authors to show that you get slow generalization when using more standard training protocols. Without this contrast, the discussion provided on p. 12 remains quite speculative (and also not especially clear).

Response: We appreciate you very much for your straight-forward excellent suggestion. While the focus of our current study is rapid generalization and its underlying mechanisms, your suggestion would allow us to test our hypothesis by contrasting rapid vs. slow development of generalization. By the way, we could be stuck by this question if without previous reviewing of our manuscript and the excellent suggestions from the reviewer 2# and you.

By following exactly your suggestion, we modified our experimental protocol to be more similar to standard training protocols for generalization study described previously, i.e. the animals were allowed to explore T-box but not G-box on the acclimation day 24 h before fear conditioning. This allowed G-box to be novel, similar, and untrained context for generalization test during retrieval. Other parts of the protocols were remained unchanged for better comparison between rapid vs. slow generalisation in our experimental conditions.

Now, the supplementary result was come out to be still exciting to us although it should be predicted by our hypothesis. The freezing level in novel G-box were only about 30% and 70% of that in T-box during the retrieval at post training 30 min and 24 h, respectively. It reached to about 90% and 98% of that in T-box during the retrieval on post training 7 d and 14 d. These results suggested that generalization development became much slower when the context memories for G-box and T-box

are not linked together before fear conditioning. In other words, fear could be transferred between the context memories more rapidly if the memories were already linked together.

We added a sentence in the method section (line 26, page 20) “For slow generalisation study, the animals were exposed to T-box but not G-box for 10 min on the acclimation day.”;

We added sentences in the discussion section (line 15, page 14) “It seems that the linked context memories would allow fear to be transferred (or generalized) from T-box to G-box rapidly within 24 h. Consistent with this assumption, without such a memory link by using the protocols similar as previous generalisation studies described through exposing the animals to T-box but not G-box on the acclimation day, full development of generalisation in G-box was observed by two weeks after fear conditioning (Supplementary Fig. 11).”

Supplementary Fig 11. | Slow generalisation of contextual fear memory in SD rat. On the acclimation day 24 h before contextual fear conditioning, rats were exposed to T-box but not G-box for 10 min. The other parts of the protocols were remained exactly the same as those described for rapid generalisation study. **a**, Independent groups of the rats ($n = 10$ for each group) were tested for contextual fear memory and generalisation at 0.5 h, and 1, 7 and 14 d after fear conditioning. The freezing level in G-box gradually rose up to a near equivalent level to that in T-box on 14 d (**Time \times box interaction, $F(3,36) = 16.557, P < 0.001$; T-box vs. G-box contrast effects, $***P < 0.001$ at 0.5 h and 1 d; $*P = 0.026$ on 7 d; $P = 0.544$ on 14 d; two-way ANOVA). **b**, The index calculated by the freezing level in G-box divided by that in T-box (G/T) indicated that generalisation development from about 30% to 98% within 14 d after fear conditioning, suggesting a time-course of slow generalisation in our experimental conditions over two weeks. This is in marked contrast to rapid generalisation that was fully developed within 24 h (see Fig. 1b).**

2. Statistical analyses. I requested that they provide more statistical details. In particular, for some of their claims they would require significant interaction terms in

their ANOVAs. These have not been provided.

Response: We apologized for not understanding your question accurately. Now, the statistical result for interaction between treatment and box was added in each result including supplementary figures wherever two-way ANOVA was used. The added details were labeled as red color in text.

We would like to thank you for your suggestion as we found that the interaction in figure 2b and 3n did not reach statistical significance:

1. In figure 2b, the interaction did not reach significance, and then we added the ANOVA for within or between subject “(Fig. 2b, group×box interaction, $F(2,34) = 2.455, P = 0.101$; within groups or between boxes, $F(2,34) = 7.933, P = 0.001$ or $F(1,34) = 29.66, P < 0.001$; two-way ANOVA; Uni: $\#P = 0.02$, G-box vs. Sham, parameter estimates, two-way ANOVA; $***P < 0.001$, G-box vs. T-box, contrast effects, two-way ANOVA)”. For this reason, we also weaken our conclusion as “might impair generalisation”.
2. In figure 3n, the interaction was not significance, and we added the ANOVA for within or between subject “(group×box interaction, $F(2,21) = 3.309, P = 0.384$; within groups or between boxes, $F(2,21) = 6.836, P = 0.005$ or $F(1,21) = 4.918, P = 0.038$; $*P = 0.017$, T-box vs. G-box, contrast effects, two-way ANOVA)”. For this reason, we also weaken our conclusion as “might be impaired by”.
3. In all other figures including supplementary figures, the interaction between treatment×box reached significance. As to laterality (left/ips and right/con CA1) in figure 2d, h, there was no significance in the interaction between treatment×box, suggesting that each side of CA1 has near equivalent contribution to generalisation impairment. The statistical details were showed in the result section.

REVIEWERS' COMMENTS:

Reviewer #3 (Remarks to the Author):

My comments/questions have been satisfactorily addressed.

Point-by-point response to the referees' comments:

REVIEWERS' COMMENTS:

Reviewer #3 (Remarks to the Author):

My comments/questions have been satisfactorily addressed.

Response: We appreciate you very much for your substantial contributions to our manuscript.